# Determinantal point process attention over grid cell code supports out of distribution generalization

**Shanka Subhra Mondal[1]\*, Steven Frankland[2], Taylor W Webb[3], Jonathan D Cohen[2]**

[1]Department of Electrical and Computer Engineering, Princeton University, Princeton, United States; [2]Princeton Neuroscience Institute, Princeton University, Princeton, United States; [3]Department of Psychology, University of California, Los Angeles, Los Angeles, United States

**\*For correspondence:** smondal@princeton.edu

**Competing interest:** The authors declare that no competing interests exist.

**Abstract** Deep neural networks have made tremendous gains in emulating human-like intelligence, and have been used increasingly as ways of understanding how the brain may solve the complex computational problems on which this relies. However, these still fall short of, and therefore fail to provide insight into how the brain supports strong forms of generalization of which humans are capable. One such case is out-of-distribution (OOD) generalization – successful performance on test examples that lie outside the distribution of the training set. Here, we identify properties of processing in the brain that may contribute to this ability. We describe a two-part algorithm that draws on specific features of neural computation to achieve OOD generalization, and provide a proof of concept by evaluating performance on two challenging cognitive tasks. First we draw on the fact that the mammalian brain represents metric spaces using grid cell code (e.g., in the entorhinal cortex): abstract representations of relational structure, organized in recurring motifs that cover the representational space. Second, we propose an attentional mechanism that operates over the grid cell code using determinantal point process (DPP), that we call DPP attention (DPP-A) – a transformation that ensures maximum sparseness in the coverage of that space. We show that a loss function that combines standard task-optimized error with DPP-A can exploit the recurring motifs in the grid cell code, and can be integrated with common architectures to achieve strong OOD generalization performance on analogy and arithmetic tasks. This provides both an interpretation of how the grid cell code in the mammalian brain may contribute to generalization performance, and at the same time a potential means for improving such capabilities in artificial neural networks.

## eLife assessment

This **important** modeling work demonstrates out-of-distribution generalization using a grid cell coding scheme combined with an attentional mechanism that operates over these representations (Determinantal Point Process Attention). The simulations provide **compelling** evidence that the model can improve generalization performance for analogies, addition, and multiplication. The paper is significant in demonstrating how neural grid codes can support human-like generalization capabilities in analogy and arithmetic tasks, which has been a challenge for prior models.

## Introduction

Deep neural networks now meet, or even exceed, human competency in many challenging task domains (*He et al., 2016*; *Silver et al., 2017*; *Wu et al., 2016*; *He et al., 2017*). Their success on these tasks, however, is generally limited to the narrow set of conditions under which they were trained,

falling short of the capacity for strong forms of generalization that is central to human intelligence (*Barrett et al., 2018*; *Lake and Baroni, 2018*; *Hill et al., 2019*; *Webb et al., 2020*), and hence fail to provide insights into how our brain supports them. One such case is out-of-distribution (OOD) generalization where the test data lies outside the distribution of the training data. Here, we consider two challenging cognitive problems that often require a capacity for OOD generalization: (1) analogy and (2) arithmetic. What enables the human brain to successfully generalize on these tasks, and how might we better realize that ability in deep learning systems?

To address the problem, we focus on two properties of processing in the brain that we hypothesize are useful for OOD generalization: (1) the *abstract representations* of relational structure, in which relations are preserved across transformations like translation and scaling (such as observed for grid cells in mammalian medial entorhinal cortex *Hafting et al., 2005*); and (2) an *attentional objective* inspired from determinantal point processes (DPPs), which are probabilistic models of repulsion arising in quantum physics (*Macchi, 1975*), to attend to abstract representations that have maximum variance and minimum correlation among them, over the training data. We refer to this as DPP attention or DPP-A. The net effect of these two properties is to normalize the representations of training and testing data in a way that preserves their relational structure, and allows the network to learn that structure in a form that can be applied well beyond the domain over which it was trained.

In previous work, it has been shown that such OOD generalization can be accomplished in a neural network by providing it with a mechanism for temporal context normalization (TCN) (*Webb et al., 2020*), a technique that allows neural networks to preserve the relational structure between the inputs in a local temporal context, while abstracting over the differences between contexts. [Temporal context normalization (*Webb et al., 2020*) is a normalization procedure proposed for use in training a neural network, similar to batch normalization (*Ioffe and Szegedy, 2015*), in which tensor normalization is applied over the temporal instead of the batch dimension, which is shown to help with OOD generalization. It is unrelated to the temporal context model (*Howard et al., 2005*), which is a computational model that proposes a role for temporal coding in the functions of the medialtemporal lobe in support of episodic recall, and spatial navigation.] Here, we test whether the same capabilities can be achieved using a well-established, biologically plausible embedding scheme – grid cell code – and an adaptive form of normalization that is based strictly on the statistics of the training data in the embedding space. We show that when deep neural networks are presented with data that exhibits such relational structure, grid cell code coupled with an error-minimizing/attentional objective promotes strong OOD generalization. We unpack each of these theoretical components in turn before describing the tasks, modeling architectures, and results.

## Abstract representations of relational structure

The first component of the proposed framework relies on the idea that a key element underlying human-like OOD generalization is the use of low-dimensional representations that emphasize the relational structure between data points. Empirical evidence suggests that, for spatial information, this is accomplished in the brain by encoding the organism's spatial position using a periodic code consisting of different frequencies and phases (akin to a Fourier transform of the space). Although grid cells were discovered for representations of space (*Hafting et al., 2005*; *Sreenivasan and Fiete, 2011*; *Mathis et al., 2012*) and used for guiding spatial behavior (*Erdem and Hasselmo, 2014*; *Bush et al., 2015*), they have since been identified in non-spatial domains, such as auditory tones (*Aronov et al., 2017*), odor (*Bao et al., 2019*), episodic memory (*Chandra et al., 2023*), and conceptual dimensions (*Constantinescu et al., 2016*). These findings suggest that the coding scheme used by grid cells may serve as a general representation of metric structure that may be exploited for reasoning about the abstract conceptual dimensions required for higher-level reasoning tasks, such as analogy and mathematics (*McNamee et al., 2022*). Of interest here, the periodic response function displayed by grid cells belonging to a particular frequency is invariant to translation by its period, and increasing the scale of a higher-frequency response gives a lower-frequency response and vice versa, making it invariant to scale across frequencies. This is particularly promising for prospects of OOD generalization: downstream systems that acquire parameters over a narrow training region may be able to successfully apply those parameters across transformations of translation or scale, given the shared structure (which can also be learned; *Cueva and Wei, 2018*; *Banino et al., 2018*; *Whittington et al., 2020*).

## DPP-A

The second component of our proposed framework is a novel attentional objective that uses the statistics of the training data to sculpt the influence of grid cells on downstream computation. Despite the use of a relational encoding metric (i.e., grid cell code), generalization may also require identifying which aspects of this encoding that could potentially be shared across training and test distributions. Here, we implement this by identifying, and restricting further processing to those grid cell embeddings that exhibit the greatest variance, but are least redundant (i.e., pairwise uncorrelated) over the training data. Formally, this is captured by maximizing the determinant of the covariance matrix of the grid cell embeddings computed over the training data (**Kulesza and Taskar, 2012**). To avoid overfitting the training data, we attend to a subset of grid cell embeddings that maximize the volume in the representational space, diminishing the influence of low-variance codes (irrelevant), or codes with high similarity to other codes (redundant), which decrease the determinant of the covariance matrix.

DPP-A is inspired by mathematical work in statistical physics using DPPs that originated for modeling the distribution of fermions at thermal equilibrium (**Macchi, 1975**). DPPs have since been adopted in machine learning for applications in which diversity in a subset of selected items is desirable, such as recommender systems (**Kulesza and Taskar, 2012**). Recent work in computational cognitive science has shown DPPs naturally capture inductive biases in human inference, such as some word-learning and reasoning tasks (e.g., one noun should only refer to one object) while also serving as an efficient memory code (**Webb et al., 2020**). In that context, the learner is biased to find a set of possible word-meaning pairs whose representations exhibit the greatest variance and lowest covariance on a task-relevant dataset. DPPs also provide a formal objective for the type of orthogonal coding that has been proposed to be characteristic of representations in mammalian hippocampus,

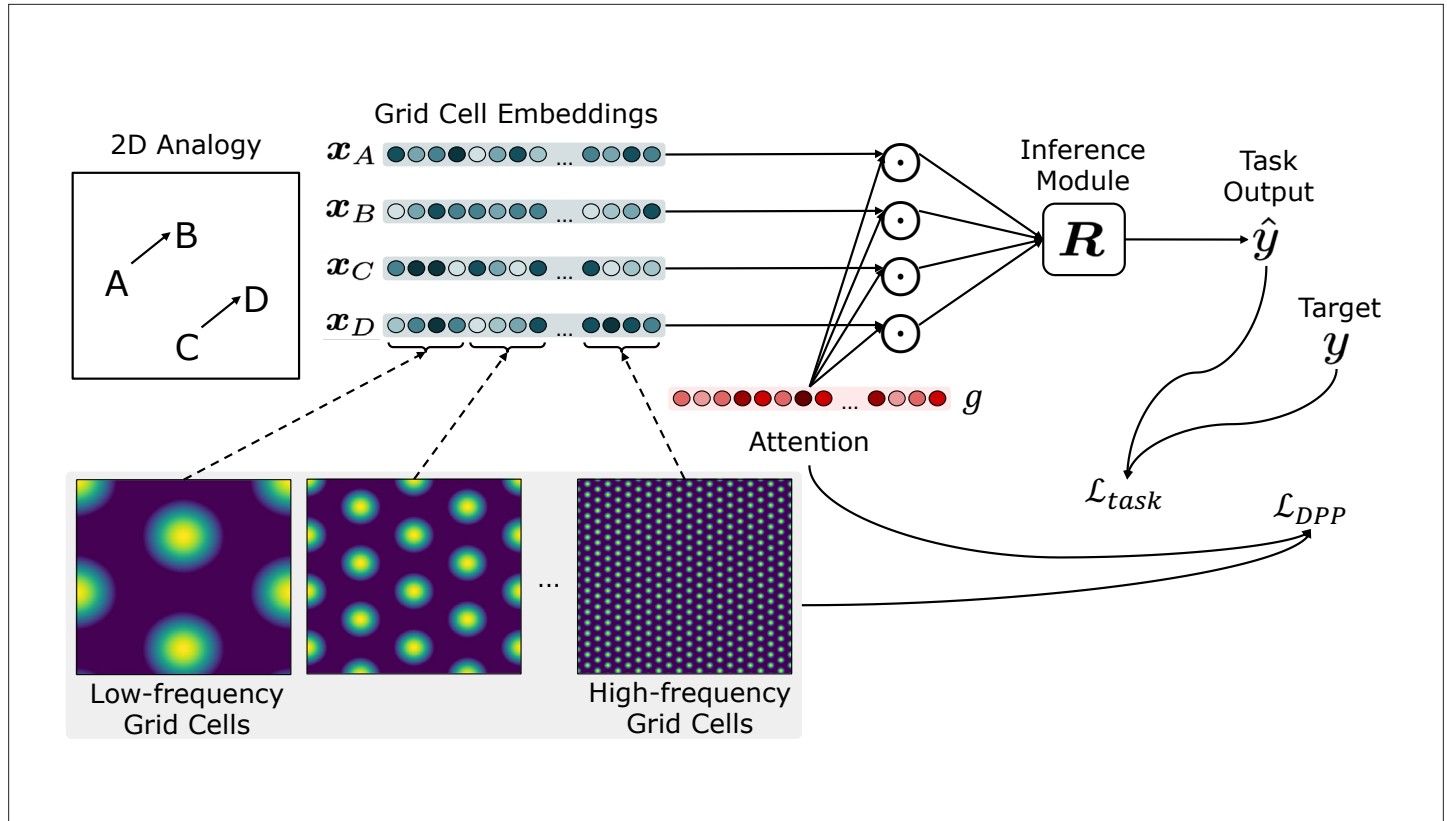

**Figure 1.** Schematic of the overall framework. Given a task (e.g., an analogy to solve), inputs (denoted as $\{A, B, C, D\}$) are represented by the grid cell code, consisting of units (grid cells) representing different combinations of frequencies and phases. Grid cell embeddings ($x_A$, $x_B$, $x_C$, $x_D$) are multiplied elementwise (represented as a Hadamard product $\odot$) by a set of learned attention gates $g$, then passed to the inference module $R$. The attention gates $g$ are optimized using $\mathcal{L}_{DPP}$, which encourages attention to grid cell embeddings that maximize the volume of the representational space. The inference module outputs a score for each candidate analogy (consisting of $A, B, C$ and a candidate answer choice $D$). The scores for all answer choices are passed through a softmax to generate an answer $\hat{y}$, which is compared against the target $y$ to generate the task loss $\mathcal{L}_{task}$.

and integral for episodic memory (*McClelland et al., 1995*). Thus, using the DPP objective to govern attention over the grid cell code, known to be implemented in the entorhinal cortex (*Hafting et al., 2005*; *Barry et al., 2007*; *Stensola et al., 2012*; *Giocomo et al., 2011*; *Brandon et al., 2011*) (one synapse upstream of the hippocampus), aligns with the function and organization of cognitive and neural systems underlying the capability for abstraction.

Taken together, the representational and attention mechanisms outlined above define a two-component framework of neural computation for OOD generalization, by minimizing task-specific error subject to: (1) embeddings that encode relational structure among the data (grid cell code), and (2) attention to those embeddings that maximize the 'volume' of the representational space that is covered, while minimizing redundancy (DPP-A). Below, we demonstrate proof of concept by showing that these mechanisms allow artificial neural networks to learn representations that support OOD generalization on two challenging cognitive tasks and therefore serve as a reasonable starting point for examining the properties of interest in these networks.

## Methods

*Figure 1* illustrates the general framework. Task inputs, corresponding to points in a metric space, are represented as a set of grid cell embeddings $x_{t=1..T}$, that are then passed to the inference module $R$. The embedding of each input is represented by the pattern of activity of grid cells that respond selectively to different combinations of phases and frequencies. Attention over these is a learned gating $g$ of the grid cells, the gated activations of which ($x \odot g$) are passed to the inference module ($R$). The parameterization of $g$ and $R$ are determined by backpropagation of the error signal obtained by two loss functions over the training set. Note that learning of parameter $g$ occurs only over the training space and is not further modified during testing (i.e., over the test spaces). The first loss function, $\mathcal{L}_{DPP}$ favors attentional gatings over the grid cells that maximize the DPP-A objective; that is, the 'volume' of the representational space covered by the attended grid cells. The second loss function, $\mathcal{L}_{task}$ is a

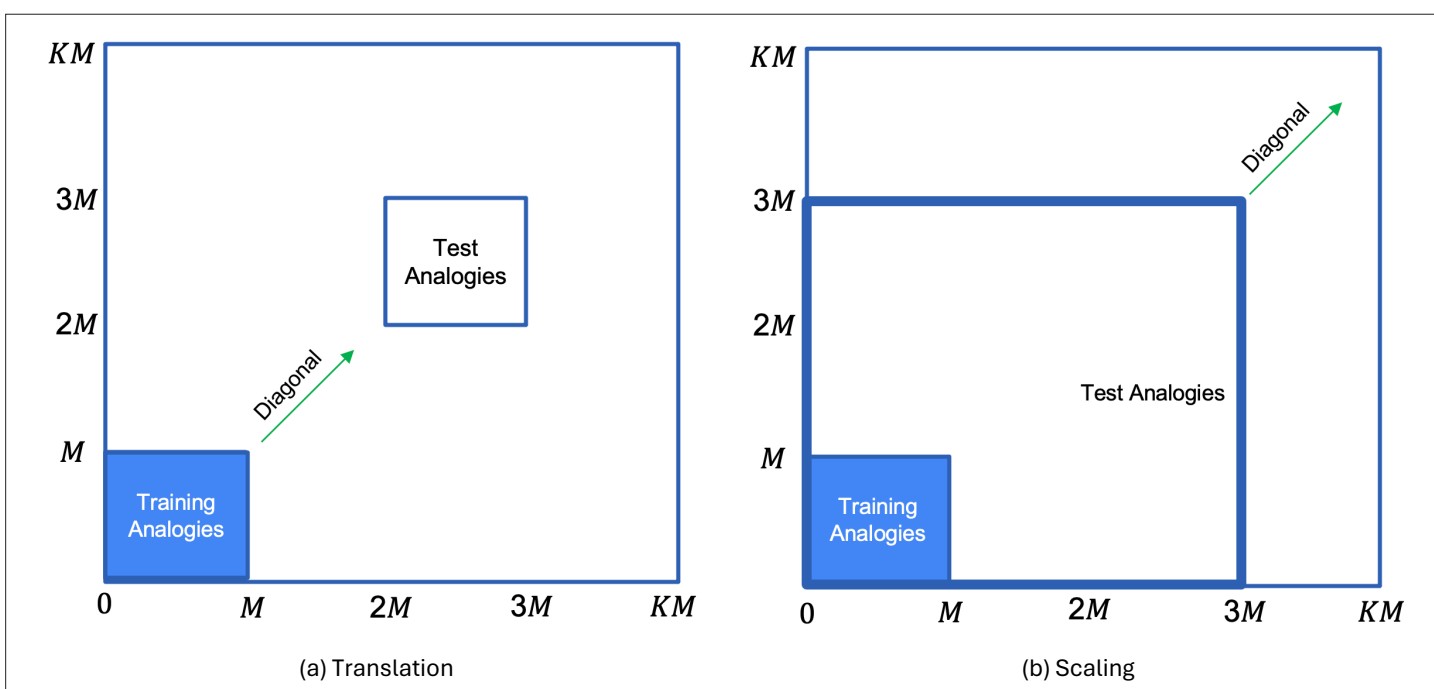

**Figure 2.** Generation of test analogies from training analogies (region marked in blue) by: (**a**) translating both dimension values of $A, B, C, D$ by the same amount; and (**b**) scaling both dimension values of $A, B, C, D$ by the same amount. Since both dimension values are transformed by the same amount, each input gets transformed along the diagonal.

The online version of this article includes the following source data for figure 2:

**Source data 1.** The zip file contains the data for the analogy task depicted in *Figure 2*.

standard task error term (e.g., the cross-entropy of targets $y$ and task outputs $\hat{y}$ over the training set). We describe each of these components in the following sections.

## Task setup
### Analogy task

We constructed proportional analogy problems with four terms, of the form $A : B : : C : D$, where the relation between $A$ and $B$ was the same as between $C$ and $D$. Each of $A, B, C, D$ was a point in the integer space $\mathbb{Z}^2$, with each dimension sampled from the range $[0, M - 1]$, where $M$ denotes the size of the training region. To form an analogy, two pairs of points $(A, B)$ and $(C, D)$ were chosen such that the vectors $AB$ and $CD$ were equal. Each analogy problem also contained a set of six foil items sampled in the range $[0, M - 1]^2$ excluding $D$, such that they did not form an analogy with $A, B, C$. The task was, given $A$, $B$, and $C$, to select $D$ from a set of multiple choices consisting of $D$ and the six foil items. During training, the networks were exposed to sets of points sampled uniformly over locations in the training range, and with pairs of points forming vectors of varying length. The network was trained on 80% of all such sets of points in the training range, with 20% held out as the validation set.

To study OOD generalization, we created two cases of test data, that tested for OOD generalization in translation and scale. For the *translation invariance* case (*Figure 2a*), the constituents of the training analogies were translated along both dimensions by the same integer value (obtained by multiplying $K$ and $M$, both of which are integer values) such that the test analogies were in the range $[KM, (K + 1) M - 1]^2$ after translation. [We transformed by the same amount along both dimensions so that the OOD generalization regimes are similar to *Webb et al., 2020*.] Non-overlapping test regions were generated for $K \in [1, 9]$. Similar to the translation OOD generalization regime of *Webb et al., 2020*, this allowed the graded evaluation of OOD generalization to a series of increasingly remote test domains as the distance from the training region increased. For example a training analogy $A : B : : C : D$ after translation by $KM$, would be $A + KM : B + KM : : C + KM : D + KM$.

For the *scale invariance* case (*Figure 2b*), we scaled each constituent of the training analogies by $K$ so that the test analogies after scaling were in the range $[0, KM - 1]^2$. Thus, an analogy $A : B : : C : D$ after scaling by $K$, would be $KA : KB : : KC : KD$. By varying the value of $K$ from 1 to 9, we scaled the training analogies to occupy increasingly distant and larger regions of the test space. It is worth noting that while humans can exhibit complex and sophisticated forms of analogical reasoning (*Holyoak, 2012*; *Lu et al., 2022*; *Webb et al., 2023*), here we focused on a relatively simple form, that was inspired by Rumelhart's parallelogram model of analogy (*Mikolov et al., 2013*; *Rumelhart and Abrahamson, 1973*) that has been used to explain traditional human verbal analogies (e.g., "king is to what as man is to woman?"). Our model, like that one, seeks to explain analogical reasoning in terms of the computation of simple Euclidean distances (i.e., $A - B = C - D$, where $A, B, C, D$ are vectors in 2D space).

### Arithmetic task

We tested two types of arithmetic operations, corresponding to the translation and scaling transformations used in the analogy tasks: elementwise addition and multiplication of two inputs $A$ and $B$, each a point in $\mathbb{Z}^2$, for which $C$ was the point corresponding to the answer (i.e., $C = A + B$ or $C = A * B$). As with the analogy task, each arithmetic problem also contained a set of six foil items sampled in the range $[0, M - 1]^2$, excluding $C$. The task was to select $C$ from a set of choices consisting of $C$ and the six foil items. Similar to the analogy task, training data was constructed from a uniform distribution of points and vector lengths in the training range, with 20% held out as the validation set. To study OOD generalization, we created testing data corresponding to $K = 9$ non-overlapping regions, such that $C \in [M, 2M - 1]^2, [2M, 3M - 1]^2, \ldots [KM, (K + 1) M - 1]^2$.

## Architecture
### Grid cell code

As discussed above, the grid cell code is found in the mammalian neocortex, that support structured, low-dimensional representations of task-relevant information. For example, an organism's location in 2D allocentric space (*Hafting et al., 2005*), the frequency of 1D auditory stimuli (*Aronov et al., 2017*), and conceptual knowledge in two continuous dimensions *Doeller et al., 2010*; *Constantinescu et al.,*

*2016* have all been shown to be represented as unique, similarity-preserving combinations of frequencies and phases. Here, these codes are of interest because the relational structure in the input is preserved in the code across translation and scale. This offers a promising metric that can be used to learn structure relevant to the processing of analogies (*Frankland et al., 2019*) and arithmetic over a restricted range of stimulus values, and then used to generalize such processing to stimuli outside of the domain of task training.

To derive the grid cell code for stimuli, we follow the analytic approach described by *Bicanski and Burgess, 2019* (https://github.com/bicanski/VisualGridsRecognitionMem; *Bicanski, 2019*). Specifically, the grid cell embedding for a particular stimulus location $A$ is given by:

$$x_A = \max\left(0, \cos(z_0) + \cos(z_1) + \cos(z_2)\right) \tag{1}$$

where,

$$z_i = b_i * \left(FA + A_{offset}\right) \tag{2}$$

$$b_0 = \begin{pmatrix} \cos(0) \\ \sin(0) \end{pmatrix}, b_1 = \begin{pmatrix} \cos\left(\frac{\pi}{3}\right) \\ \sin\left(\frac{\pi}{3}\right) \end{pmatrix}, b_2 = \begin{pmatrix} \cos\left(\frac{2\pi}{3}\right) \\ \sin\left(\frac{2\pi}{3}\right) \end{pmatrix} \tag{3}$$

The spatial frequencies of grids ($F$) begin at a value of $0.0028 * 2\pi$. *Wei et al., 2015* have shown that, to minimize the number of variables needed to represent an integer domain of size $S$, the firing rate widths and frequencies should scale geometrically in a range ($\sqrt{2}, \sqrt{e}$), closely matching empirically observed scaling in entorhinal cortex (*Stensola et al., 2012*). We choose a scaling factor of $\sqrt{2}$ to efficiently tile the space. One consequence of this efficiency is that the total number of discrete frequencies in the entorhinal cortex is expected to be small. Empirically, it has been estimated to be between 8 and 12 (*Moser et al., 2015*) [It seems likely that the use of grid cell code for abstraction in human cognition requires a considerably greater number of states $S$ than that used by the rodent for sensory encoding. However, given exponential scaling, the total number of frequencies is expected to remain low, increasing as a logarithm of $S$.]. Here, we choose $N_f = 9$ (dimension of $F$) as the number of frequencies. $A$ refers to a particular location in a two dimensional space, and 100 offsets ($A_{offset}$) are used for each frequency to evenly cover a space of 1000 × 1000 locations using 900 grid cells. These offsets represent different phases within each frequency and since there are 100 of them, $N_p = 100$. Hence $N_p \times N_f = 900$, which denotes the number of grid cells. Each point from the set of 2D points for the stimuli in a task (described in Task setup), was represented using the firing rate of 900 grid cells which constituted the grid cell embedding for that point to form the inputs to our model.

## DPP-A

We hypothesize that the use of a relational encoding metric (i.e., grid cell code) is extremely useful, but not sufficient for a system to achieve strong generalization, which requires attending to particular aspects of the encoding that can capture the same relational structure across the training and test distributions. Toward this end, we propose an attentional objective that uses the statistics of the training data to attend to grid cell embeddings that can induce the inference module to achieve strong generalization. Our objective, which we describe in detail below, seeks to identify those grid cell embeddings that exhibit the greatest variance but are least redundant (pairwise uncorrelated over the training data). Formally, this is captured by maximizing the determinant of the covariance matrix of the grid cell embeddings computed over the training data (*Kulesza and Taskar, 2012*). Although in machine learning, DPPs have been particularly influential in work on recommender systems (*Chen et al., 2018*), summarization (*Gong et al., 2014*; *Perez-Beltrachini and Lapata, 2021*), and neural network pruning (*Mariet and Sra, 2015*), here, we propose to use maximization of the determinant of the covariance matrix as an attentional mechanism to limit the influence of grid cell embeddings with low-variance (which are less relevant) or with high similarity to other grid cell embeddings (which are redundant).

For the specific tasks that we study here, we have assumed the grid cell embeddings to be pre-learned to represent the entire space of possible test data points, and we are simply focused on the problem of how to determine which of these are most useful in enabling generalization for a

task-optimized network trained on a small fraction of that space (*Figure 2*). That is, we look for a way to attend to a subset of grid cells frequencies whose embeddings capture recurring task-relevant relational structure. We find that grid cell embeddings corresponding to the higher spatial frequency grid cells exhibit greater variance over the training data than the lower-frequency embeddings, while at the same time the correlations among those grid cell embeddings are lower than the correlations among the lower-frequency grid cell embeddings. The determinant of the covariance matrix of the grid cell embeddings is maximized when the variances of the grid cell embeddings are high (they are 'expressive') and the correlation among the grid cell embeddings is low (they 'cover the representational space'). As a result, the higher-frequency grid cell embeddings more efficiently cover the representational space of the training data, allowing them to efficiently capture the same relational structure across training and test distributions which is required for OOD generalization.

---

**Algorithm 1. Training with DPP-A**

---

Parameters: inference module $R$, attention gates $g$
Hyperparameters: number of frequencies $N_f$, number of phases $N_p$, number of epochs optimizing DPP attention $N_{E_{DPP}}$, number of epochs optimizing task loss $N_{E_{task}}$, number of batches per epoch $N_b$
Inputs: covariance matrix $V$, grid cell embeddings $x$ and targets $y$ for all training problems

---

Initialize $g$, $R$
for $i = 1$ to $N_{E_{DPP}}$ do
 for $j = 1$ to $N_b$ do

$$\hat{F}\left(g, V, N_f, N_p\right) = \sum_{f=1}^{N_f} \log \det \left(\text{diag}\left(\sigma(g_f)\right)\left(V_f - I\right) + I\right)$$

 $\mathcal{L}_{DPP} = -\hat{F}(g, V, N_f, N_p)$
 Update $g$
 end for
end for
$\hat{F}_{f \in [1, N_f]} = \log \det \left(\text{diag}\left(\sigma(g_f)\right)\left(V_f - I\right) + I\right)$
$f_{max_{DPP}} = \arg\max_{f \in [1, N_f]} \hat{F}_f$
for $i = 1$ to $N_{E_{task}}$ do
 for $j = 1$ to $N_b$ do
 $\hat{y} = R(x_{f=f_{max_{DPP}}})$
 $\mathcal{L}_{task} = \text{cross-entropy}(\hat{y}, y)$
 Update $R$
 end for
end for

---

Formally, we treat obtaining $\mathcal{L}_{DPP}$ as an approximation of a DPP. A DPP $\mathcal{P}$ defines a probability measure on all subsets of a set of items $\mathcal{U} = \{1, 2, \dots N\}$. For every $u \subseteq \mathcal{U}$, $P(u) \propto \det(V_u)$. Here, $V$ is a positive semidefinite covariance matrix and $V_u = [V_{ij}]_{i,j \in u}$ denotes the matrix $V$ restricted to the entries indexed by elements of $u$. The maximum a posteriori (MAP) problem $argmax_u \det(V_u)$ is NP-hard (*Ko et al., 1995*). However, $f(u) = \log\left(\det(V_u)\right)$ satisfies the property of a submodular function, and various algorithms exist for approximately maximizing them. One common way is to approximate this discrete optimization problem by replacing the discrete variables with continuous variables and extending the objective function to the continuous domain. *Gillenwater et al., 2012* proposed a continuous extension that is efficiently computable and differentiable:

$$\hat{F}(w) = \log \sum_{u} \prod_{i \in u} w_i \prod_{i \in u} (1 - w_i) \exp\left(f(u)\right), w \in [0, 1]^N. \tag{4}$$

We use the following theorem from *Gillenwater et al., 2012* to construct $\mathcal{L}_{DPP}$:

## Theorem 2.1
For a positive semidefinite matrix $V$ and $w \in [0, 1]^N$:

$$\sum_{u} \prod_{i \in u} w_i \prod_{i \in u} (1 - w_i) \det(V_u) = \det\left(\text{diag}(w)(V - I) + I\right) \tag{5}$$

We propose an attention mechanism that uses $\mathcal{L}_{DPP}$ to attend to subsets of grid cell embeddings for further processing. Algorithm 1 describes the training procedure with DPP-A which consists of two steps, using $\mathcal{L}_{DPP}$ as the first step. This step maximizes the objective function:

$$\hat{F}\left(\boldsymbol{g}, \boldsymbol{V}, N_f, N_p\right) = \sum_{f=1}^{N_f} \log \det \left(\text{diag}\left(\sigma(\boldsymbol{g}_f)\right)\left(\boldsymbol{V}_f - \boldsymbol{I}\right) + \boldsymbol{I}\right) \tag{6}$$

using stochastic gradient ascent for $N_{E_{DPP}}$ epochs, which is equivalent to minimizing $\mathcal{L}_{DPP}$, as $\mathcal{L}_{DPP} = -\hat{F}(\boldsymbol{g}, \boldsymbol{V}, N_f, N_p)$. It involves attending to grid cell embeddings that exhibit the greatest within-frequency variance but are least redundant (i.e., that are least also pairwise uncorrelated) over the training data. This is achieved by maximizing the determinant of the covariance matrix over the within-frequency grid cell embeddings of the training data, and *Equation 6* is obtained by applying log on both sides of the Theorem 2.1, and in our case $\mathcal{U}$ refers to grid cells of a particular frequency. Here, $\boldsymbol{g}$ are the attention gates corresponding to each grid cell, and $N_f$ is the number of distinct frequencies. The matrix $\boldsymbol{V}$ captures a measure of the covariance of the grid cell embeddings over the training region. We used the *synth_kernel* function (https://github.com/insuhan/fastdppmap/blob/db7a28c3 8ce654bdbfd5ab1128d3d5910b68df6b/test_greedy.m#L123; *Han, 2017*) to construct $\boldsymbol{V}$, where in our case $\boldsymbol{m}$ are the variances of the grid cell embeddings $S$ computed over the training region $M$, $N$ is the number of grid cells and $w_m, b$ are hyperparameters with values of 1 and 0.1, respectively. [$S$ need not be a square matrix in our case, whose second dimension $M$ is the size of the training region. $L\_kernel$ is same as $\boldsymbol{V}$.] The dimensionality of $\boldsymbol{V}$ is $N_f N_p \times N_f N_p$ ($900 \times 900$). $\boldsymbol{g}_f$ are the gates of the grid cells belonging to the $f$th frequency, so $\boldsymbol{g}_f = \boldsymbol{g}[f N_p : (f+1)N_p]$, where $N_p$ is the number of phases for each frequency. $\boldsymbol{V}_f = \boldsymbol{V}[f N_p : (f+1)N_p, f N_p : (f+1)N_p]$ is the restriction of $\boldsymbol{V}$ to the grid cell embeddings for $f$ th frequency, so it captured the covariance of the grid cell embeddings belonging to the $f$ th frequency. $\sigma$ is sigmoid nonlinearity applied to $\boldsymbol{g}_f$, defined as $\sigma(\boldsymbol{g}_f) = 1/(1 + e^{-\boldsymbol{g}_f})$, so that their values are between 0 and 1. $\text{diag}(\sigma(\boldsymbol{g}_f))$ converts vector $\sigma(\boldsymbol{g}_f)$ into a matrix with $\sigma(\boldsymbol{g}_f)$ as the diagonal of the matrix and the rest entries are zero, which is multiplied with $\boldsymbol{V} - \boldsymbol{I}$, which results in elementwise multiplication of $\sigma(\boldsymbol{g}_f)$ with the column vectors of $\boldsymbol{V} - \boldsymbol{I}$. *Equation 6* which involved summation of the logarithm of the determinant of the gated covariance matrix of grid cell embeddings within each frequency, over $N_f$ frequencies was used to compute the negative of $\mathcal{L}_{DPP}$. Maximizing $\hat{F}$ gave the approximate maximum within-frequency log determinant for each frequency $f \in [1, N_f]$, which we denote for the $f$ th frequency as $\hat{F}_f$. In the second step of the training procedure, we used the $f_{max_{DPP}}$ grid cell frequency, where $f_{max_{DPP}} = \arg\max_{f \in [1, N_f]} \hat{F}_f$. In other words, we used the grid cell embeddings for the frequency which had the maximum within-frequency log determinant at the end of the first step, which we find are best at capturing the relational structure across the training and testing data, thereby promoting OOD generalization. In this step, we trained the inference module minimizing $\mathcal{L}_{task}$ over $N_{E_{task}}$ epochs. More details can be found in Appendix 1: 'DPP-A attentional modulation'.

## Inference module

We implemented the inference module $\boldsymbol{R}$ in two forms, one using Long Short Term Memory (LSTM) (*Hochreiter and Schmidhuber, 1997*) and the other using a transformer (*Vaswani et al., 2017*) architecture. Separate networks were trained for the analogy and arithmetic tasks using each form of inference module. For each task, the attended grid cell embeddings of each stimulus obtained from the DPP-A process ($x_{f=f_{max_{DPP}}}$), were provided to $\boldsymbol{R}$ as its inputs, which we denote as $\boldsymbol{x_R}$ for brevity. For the arithmetic task, we also concatenated to $\boldsymbol{x_R}$ a one-hot tensor of dimension 2, before feeding to $\boldsymbol{R}$ that specified which computation to perform (addition or multiplication). As proposed by *Hill et al., 2019*, we treated both the analogy and arithmetic tasks as scoring (i.e., multiple choice) problems. For each analogy, the inference module was presented with multiple problems, each consisting of three stimuli, $A, B, C$, and a candidate completion from the set containing $D$ (the correct completion) and six foil completions. For each instance of the arithmetic task, it was presented with two stimuli, $A, B$, and a candidate completion from the set containing $C$ (the correct completion) and six foil completions. Stimuli were presented sequentially for $\boldsymbol{R}$ constructed using an LSTM, which consists of three gates, and computations using them defined as below:

$$\boldsymbol{input\_gate}\left[t\right] = \sigma\left(\boldsymbol{W_{ii}x_R}\left[t\right] + \boldsymbol{b_{ii}} + \boldsymbol{W_{hi}h}\left[t-1\right] + \boldsymbol{b_{hi}}\right) \tag{7}$$

$$forget\_gate\,[t] = \sigma\left(W_{if}x_R\,[t] + b_{if} + W_{hf}h\,[t-1] + b_{hf}\right) \tag{8}$$

$$output\_gate\,[t] = \sigma\left(W_{io}x_R\,[t] + b_{io} + W_{ho}h\,[t-1] + b_{ho}\right) \tag{9}$$

$$c\,[t] = forget\_gate\,[t] \odot c\,[t-1] + input\_gate\,[t] \odot \tanh\left(W_{ic}x_R\,[t] + b_{ic} + W_{hc}h\,[t-1] + b_{hc}\right) \tag{10}$$

$$h\,[t] = output\_gate\,[t] \odot \tanh\left(c\,[t]\right) \tag{11}$$

where $W_{ii}, W_{hi}, W_{if}, W_{hf}, W_{io}, W_{ho}, W_{ic}, W_{hc}$ are weight matrices and $b_{ii}, b_{hi}, b_{if}, b_{hf}, b_{io}, b_{ho}, b_{ic}, b_{hc}$ are bias vectors. $h[t]$ and $c[t]$ are the hidden state and the cell state at time $t$, respectively. The hidden state for the last timestep was passed through a linear layer with a single output unit to generate a score for the candidate completions for each problem. We used a single layered LSTM of 512 hidden units, which corresponds to the size of the hidden state, cell state, bias vectors, and weight matrices. The hidden and cell state of the LSTM was reinitialized at the start of each sequence for each candidate completion.

For $R$ constructed using a transformer, we used the standard multi-head self attention (*MHSA*) mechanism followed by a multi-layered perceptron (*MLP*) – a feedforward neural network with one hidden layer, with layer normalization (*Ba et al., 2016*) (*LN*) to transform $x_R$ at each layer of the transformer, defined as below:

$$SA\left(Q, K, V\right) = softmax\left(\frac{QK^T}{\sqrt{d_k}}\right)V \tag{12}$$

$$MHSA(x_R) = Concat\left(SA\left(W_1^Q x_R, W_1^K x_R, W_1^V x_R\right), ...., SA\left(W_H^Q x_R, W_H^K x_R, W_H^V x_R\right)\right)W^O \tag{13}$$

$$\tilde{x}_R = MLP\left(LN\left(MHSA\left(LN\left(x_R\right)\right) + x_R\right)\right) + MHSA\left(LN\left(x_R\right)\right) + x_R \tag{14}$$

where $Q$, $K$, and $V$ are called the query, key, and value matrices, respectively, $d_k$ is the dimension of the matrices, $W^Q$, $W^K$, and $W^V$ are the corresponding projection matrices, $W^O$ is the output projection matrix which is applied to the concatenation of self attention (*SA*) output for each head, and $H$ is the number of heads. The *softmax* function is used to convert real valued vector inputs into a probability distribution, defined as $softmax(z_i) = e^{z_i} / \sum_i e^{z_i}$. We used a transformer with 6 layers, each of which had 8 heads, $d_k = 32$, and *MLP* hidden layer dimension of 512. The stimuli were presented together and projected into 128 dimensions to be more easily divisible by the number of heads, followed by layer normalization. Since a transformer is naturally invariant to the order of the stimuli, to make use of the order we added to $x_R$ a learnable positional encoding (*Kazemnejad, 2019*), which is the linear projection of one-hot tensors denoting the position of stimuli in the sequence. We then concatenated a learned CLS (short for 'classification') token (analogous to the CLS token in *Devlin et al., 2018*) to $x_R$, before transforming with *Equation 14*. We took the transformed value ($\tilde{x}_R$) corresponding to the CLS token, and passed it to a linear layer with 1 output unit to generate a score for each candidate completion. This procedure was repeated for each candidate completion.

The seven scores (one for the correct completion and for six foil completions) were normalized using a softmax function, such that a higher score would correspond to a higher probability and vice versa, and the probabilities sum to 1. The inference module was trained using the task-specific cross-entropy loss ($\mathcal{L}_{task}$ = cross-entropy) between the softmax-normalized scores and the index for the correct completion (target), defined as $-\log(softmax(\text{scores})[\text{target}])$. It is worth noting that the properties of equivariance hold, since the probability distribution after applying softmax remains the same when the transformation (translation or scaling) is applied to the scores for each of the answer choices obtained from the output of the inference module, and when the same transformation is applied to the stimuli for the task and all the answer choices before presenting as input to the inference module to obtain the scores. We also tried formulating the tasks as regression problems, the details of which can be found in Appendix 1: 'Regression formulation'.

While our model is not construed to be as specific about the implementation of the $R$ module, we assume that – as a standard deep learning component – it is likely to map onto neocortical structures that interact with the entorhinal cortex and, in particular, regions of the prefrontal–posterior parietal network widely believed to be involved in abstract relational processes (*Waltz et al., 1999*; *Christoff et al., 2001*; *Knowlton et al., 2012*; *Summerfield et al., 2020*). In particular, the role of the prefrontal cortex in the encoding and active maintenance of abstract information needed for task performance (such as rules and relations) has often been modeled using gated recurrent networks,

such as LSTMs (**Frank et al., 2001**; **Braver and Cohen, 2000**), and the posterior parietal cortex has long been known to support 'maps' that may provide an important substrate for computing complex relations (**Summerfield et al., 2020**).

## Experiments

### Experimental details

For each task, the sequence of stimuli for a given problem was encoded using grid cell code (see Grid cell code), that were then modulated by DPP-A (see DPP-A), and passed to the inference module $R$ (see Inference module). We trained three networks using each type of inference module. For networks using an LSTM in the inference module, we trained each network for number of epochs for optimizing DPP attention $N_{E_{DPP}} = 50$, number of epochs for optimizing task loss $N_{E_{task}} = 50$, on analogy problems, and for $N_{E_{DPP}} = 500$, $N_{E_{task}} = 500$, on arithmetic problems with a batch size of 256, using the ADAM optimizer (**Kingma and Ba, 2014**), and a learning rate of $1e^{-3}$. For networks using a transformer in the inference module, we trained with a batch size of 128 on analogy with a learning rate of $5e^{-4}$, and on arithmetic problems with a learning rate of $5e^{-5}$. More details can be found in Appendix 1: 'More experimental details'.

### Comparison models

To evaluate how the grid cell code coupled with DPP-A compares with other architectures and approaches to generalization, and the extent to which each of these components contributed to the performance of the model, we compared it with several alternative models for performing the analogy and arithmetic tasks. First, we compared it with the TCN model (**Webb et al., 2020**) (see Related work), but modified so as to use the grid cell code as input. We passed the grid cell embeddings for each input through a shared feedforward encoder which consisted of two fully connected layers with 256 units per layer. ReLU nonlinearities were used in both the layers. The final embedding was generated with a linear layer of 256 units. TCN was applied to these embeddings and then passed as a sequence for each candidate completion to the inference module. This implementation of TCN involved a learned encoder on top of the grid cell embeddings, so it is closely analogous to the original TCN.

Next, we compared our model to one that used variational dropout (**Gal and Ghahramani, 2016**), which is shown to be more effective in generalization compared to naive dropout (**Srivastava et al., 2014**). We randomly sampled a dropout mask (50% dropout), zeroing out the contribution of some of the grid cell code in the input to the inference module. We then use that locked dropout mask for every time step in the sequence. We also compared DPP-A to a model that had an additional penalty (L1 regularization) proportional to the absolute sum of the attention gates $g$ along with the task-specific loss. We experimented with different values of $\lambda$, which controlled the strength of the penalty relative to the cross-entropy loss. We report accuracy values for $\lambda$ that achieved the best performance on the validation set. Accuracy values for various $\lambda$s are provided in Appendix 1: 'Effect of L1 Regularization strength ($\lambda$)'. Dropout and L1 regularization were chosen as a proxy for DPP-A and hence we used the same input data for fair comparison. Finally, we compared to a model that used the complete grid cell code, that is no DPP-A.

## Related work

A body of recent computational work has shown that representations similar to grid cells can be derived using standard analytical techniques for dimensionality reduction (**Dordek et al., 2016**; **Stachenfeld et al., 2017**), as well as from error-driven learning paradigms (**Cueva and Wei, 2018**; **Banino et al., 2018**; **Whittington et al., 2020**; **Sorscher et al., 2023**). Previous work has also shown that grid cells emerge in networks trained to generalize to novel location/object combinations, and support transitive inference (**Whittington et al., 2020**). Here, we show that grid cells enable strong OOD generalization when coupled with the appropriate attentional mechanism. Our proposed method is thus complementary to these previous approaches for obtaining grid cell representations from raw data.

In the field of machine learning, DPPs have been used for supervised video summarization (**Gong et al., 2014**), diverse recommendations (**Chen et al., 2018**), selecting a subset of diverse neurons to prune a neural network without hurting performance (**Mariet and Sra, 2015**), and diverse mini-batch attention for stochastic gradient descent (**Zhang et al., 2017**). Recently, **Mariet et al., 2019**

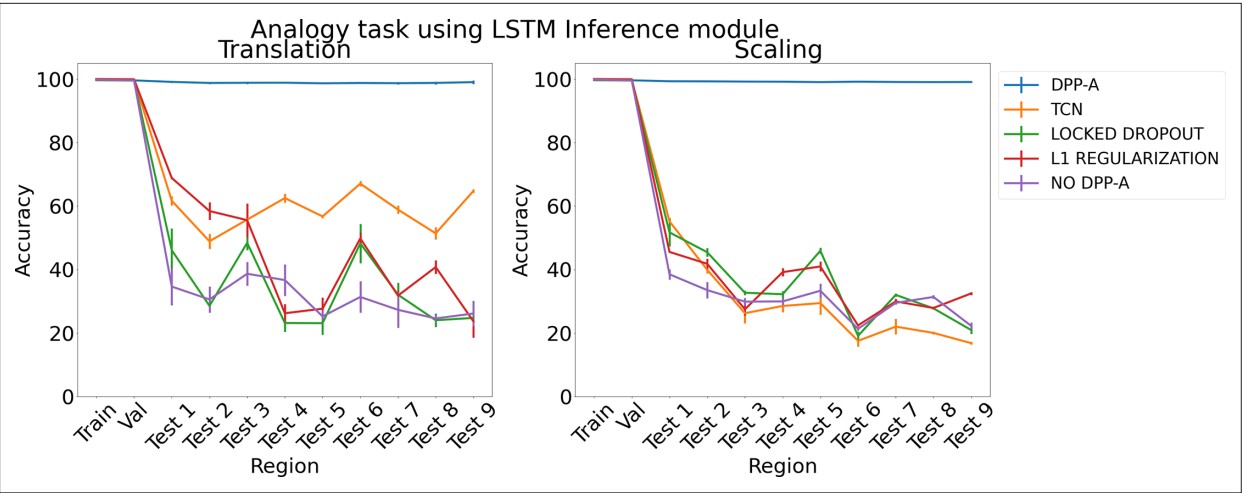

**Figure 3.** Results on analogy on each region for translation and scaling using LSTM in the inference module.

generated approximate DPP samples by proposing an inhibitive attention mechanism based on transformer networks as a proxy for capturing the dissimilarity between feature vectors, and **Perez-Beltrachini and Lapata, 2021** used DPP-based attention with seq-to-seq architectures for diverse and relevant multi-document summarization. To our knowledge, however, DPPs have not previously been combined with the grid cell code that we employ here, and have not been used to enable OOD generalization.

Various approaches have been proposed to prevent deep learning systems from overfitting, and enable them to generalize. A commonly employed technique is weight decay (**Krogh and Hertz, 1992**). **Srivastava et al., 2014** proposed dropout, a regularization technique which reduces overfitting by randomly zeroing units from the neural network during training. Recently, **Webb et al., 2020** proposed TCN in which a normalization similar to batch normalization (**Ioffe and Szegedy, 2015**) was applied over the temporal dimension instead of the batch dimension. However, unlike these previous approaches, the method reported here achieves nearly perfect OOD generalization when operating over the appropriate representation, as we show in the results. Our proposed method also has the virtue of being based on a well understood, and biologically plausible, encoding scheme (grid cell code).

## Results

### Analogy

We first present results on the analogy task for two types of testing data, translation and scaling using two types of inference module, LSTM and transformer. We trained three networks for each method and report mean accuracy along with standard error of the mean. **Figure 3** shows the results for the analogy task using an LSTM in the inference module. The left panel shows results for the translation regime, and the right panel shows results for the scaling regime. Both plots show accuracy on the training and validation sets, and on a series of nine (increasingly distant) OOD generalization test regions. DPP-A (shown in blue) achieves nearly perfect accuracy on all of the test regions, considerably outperforming the other models.

For the case of translation, using all the grid cell code with no DPP-A (shown in purple) led to the worst OOD generalization performance, overfitting on the training set. Locked dropout (denoted by green) and L1 regularization (denoted by red) reduced overfitting and demonstrated better OOD generalization performance than no DPP-A but still performed considerably worse than DPP-A. For the case of scaling, locked dropout and L1 regularization performed slightly better than TCN, achieving marginally higher test accuracy, but DPP-A still substantially outperformed all other models, with a nearly 70% improvement in average test accuracy.

To test the generality of the grid cell code and DPP-A across network architectures, we also tested a transformer (**Vaswani et al., 2017**) in place of the LSTM in the inference module. Previous work has

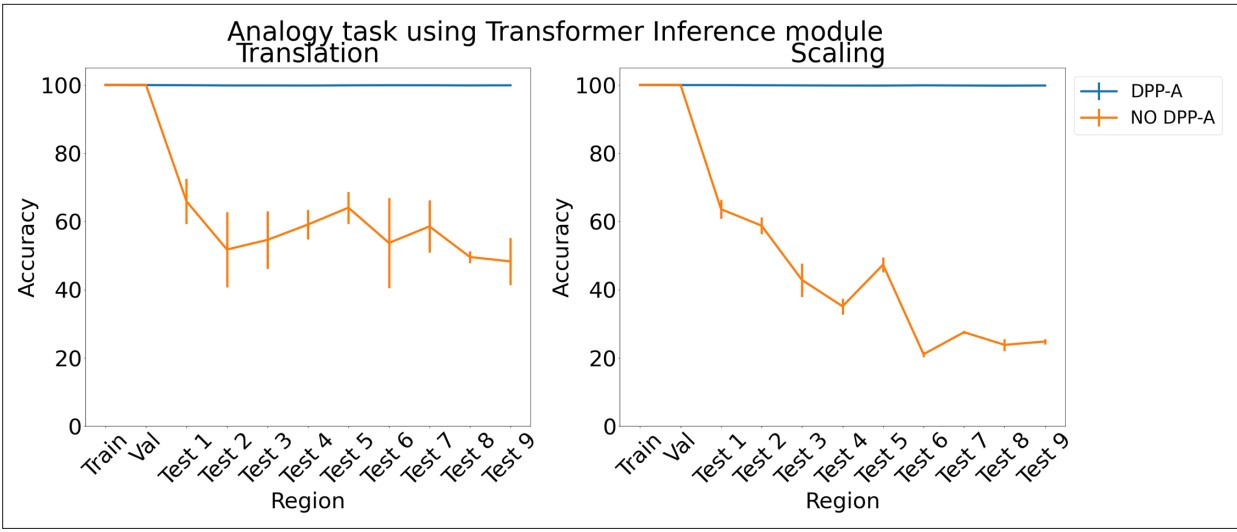

**Figure 4.** Results on analogy on each region for translation and scaling using the transformer in the inference module.

suggested that transformers are particularly useful for extracting structure in sequential data and have been used for OOD generalization (*Saxton et al., 2019*). *Figure 4* shows the results for the analogy task using a transformer in the inference module. With no explicit attention (no DPP-A) over the grid cell code (shown in orange), the transformer did poorly on the analogies on the test regions. This suggests that simply using a more sophisticated architecture with standard forms of attention is not sufficient to enable OOD generalization based on the grid cell code. With DPP-A (shown in blue), the OOD generalization performance of the transformer is nearly perfect for both translation and scaling. These results also demonstrate that grid cell code coupled with DPP-A can be exploited for OOD generalization effectively by different architectures.

## Arithmetic

We next present results on the arithmetic task for two types of operations, addition and multiplication using two types of inference module, LSTM and transformer. We trained three networks for each method and report mean accuracy along with standard error of the mean.

*Figure 5* shows the results for arithmetic problems using an LSTM in the inference module. The left panel shows results for addition problems, and the right panel shows results for multiplication problems. DPP-A achieves higher accuracy for addition than multiplication on the test regions. However, in both cases DPP-A significantly outperforms the other models, achieving nearly perfect OOD

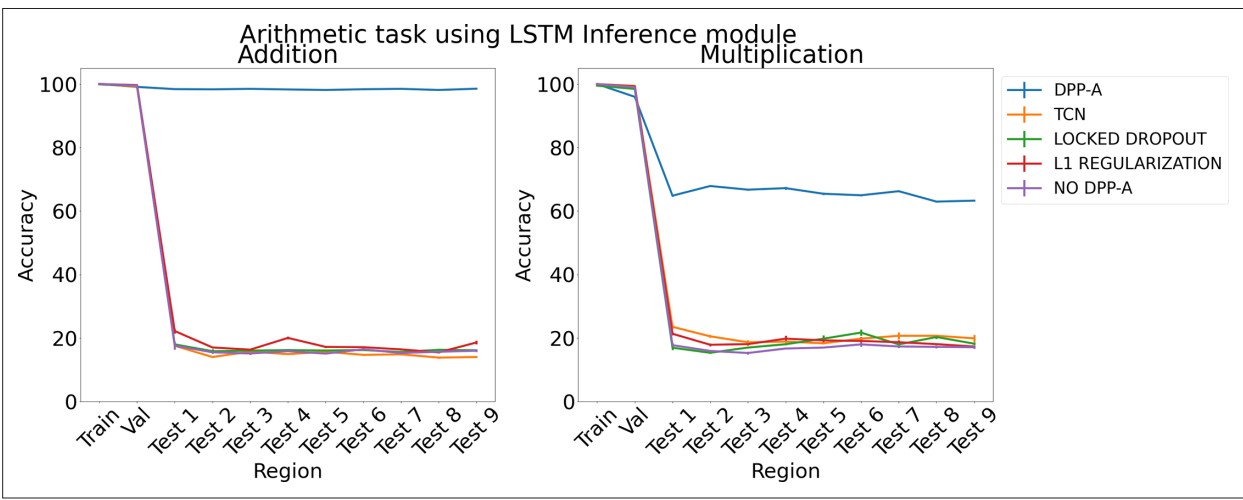

**Figure 5.** Results on arithmetic on each region using LSTM in the inference module.

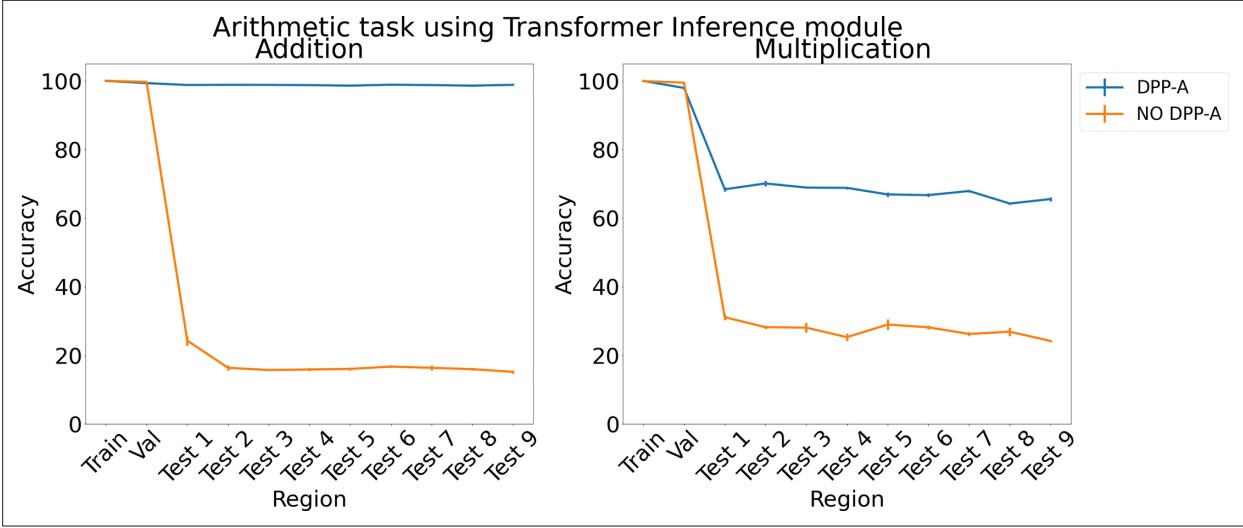

**Figure 6.** Results on arithmetic on each region using the transformer in the inference module.

generalization for addition, and 65% accuracy for multiplication, compared with under 20% accuracy for all the other models. We found that grid cell embeddings obtained after the first step in Algorithm 1 are not able to fully preserve the relational structure for multiplication problems on the test regions (more details in Appendix 1: 'Why is OOD generalization performance worse for the multiplication task?'), but still it affords superior capacity for OOD generalization than any of the other models. Thus, while it does not match the generalizability of a genuine algorithmic (i.e., symbolic) arithmetic function, it may be sufficient for some tasks (e.g., approximate multiplication ability in young children; *Qu et al., 2021*).

*Figure 6* shows the results for arithmetic problems using a transformer in the inference module. With no DPP-A over the grid cell code the transformer did poorly on addition and multiplication on the test regions, achieving around 20–30% accuracy. With DPP-A, the OOD generalization performance of the transformer shows a pattern similar to that for LSTM: it is nearly perfect for addition and, though not as good on multiplication, nevertheless shows approximately 40% better performance than the transformer multiplication.

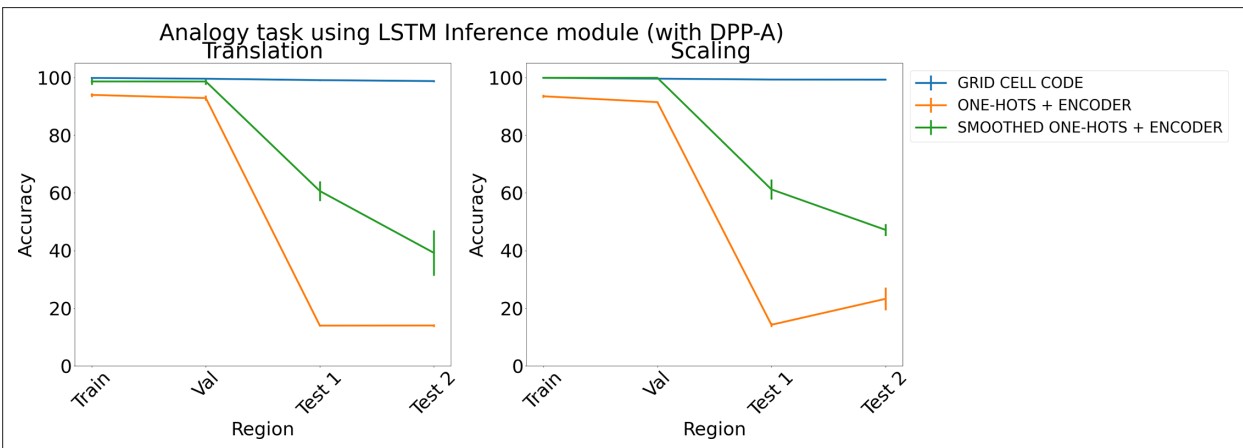

**Figure 7.** Results on analogy on each region using determinantal point process attention (DPP-A), an LSTM in the inference module, and different embeddings (grid cell code, one-hots, and smoothed one-hots passed through a learned encoder) for translation (left) and scaling (right). Each point is mean accuracy over three networks, and bars show standard error of the mean.

## Ablation study

To determine the individual importance of the grid cell code and the DPP-A attention objective, we carried out several ablation studies. First, to evaluate the importance of grid cell code, we analyzed the effect of DPP-A with other embeddings, using either one-hot or smoothed one-hot embeddings (similar to place cell coding) with standard deviations of 1, 10, and 100, each passed through a learned feedforward encoder, which consisted of two fully connected layers with 1024 units per layer, and ReLU nonlinearities. The final embedding was generated with a fully connected layer with 1024 units and sigmoid nonlinearity. Since these embeddings do not have a frequency component, the training procedure with DPP-A consisted of only one step: minimizing the loss function $\mathcal{L} = \mathcal{L}_{task} - \lambda * \hat{F}(\boldsymbol{g}, \boldsymbol{V})$. We tried different values of $\lambda$ (0.001, 0.01, 0.1, 1, 10, 100, 1000, 10,000). For each type of embedding (one-hots and smoothed one-hots with each value of standard deviation), we trained three networks and report for the model that achieved best performance on the validation set. Note that, given the much higher dimensionality and therefore memory demands of embeddings based on one-hots and smoothed one-hots, we had to limit the evaluation to a subset of the total space, resulting in evaluation on only two test regions (i.e., $K \in [1, 3]$).

*Figure 7* shows the results for the analogy task (results for the arithmetic task are in Appendix 1: 'Ablation study on arithmetic task', *Appendix 1—figure 3*) using an LSTM in the inference module. The average accuracy on the test regions for translation and scaling using smoothed one-hots passed through an encoder (shown in green) is nearly 30% better than simple one-hot embeddings passed through an encoder (shown in orange), but both still achieve significantly lower test accuracy than grid cell code which support perfect OOD generalization.

With respect to the importance of the DPP-A, we note that the simulations reported earlier show that replacing the DPP-A mechanism with either other forms of regularization (dropout and L1 regularization; see Comparison models) or a transformer (*Figure 4* in Analogy for analogy and *Figure 6* in Arithmetic for arithmetic tasks) failed to achieve the same level of OOD generalization as the network that used DPP-A. The results using a transformer are particularly instructive, as that incorporates a powerful mechanism for learned attention, but, even when provided with grid cell embeddings, failed to produce results comparable to DPP-A, suggesting that the latter provides a simple but powerful form of attention objective, at least when used in conjunction with grid cell embeddings.

Finally, for completeness, we also carried out a set of simulations that examined the performance of networks with various embeddings (grid cell code, and one-hots or smoothed one-hots with or without a learned feedforward encoder), but no attention or regularization (i.e., neither DPP-A, transformer, nor TCN, L1 Regularization, or Dropout). *Figure 8* shows the results for the different embeddings on the analogy task (results for the arithmetic task are in Appendix 1: 'Ablation study on arithmetic task', *Appendix 1—figure 4*). For translation (left), the average accuracy over the test regions using grid cell code (shown in blue) is nearly 25% more compared to other embeddings, which all yield performance near chance (~15%). For scaling (right), although other embeddings achieve

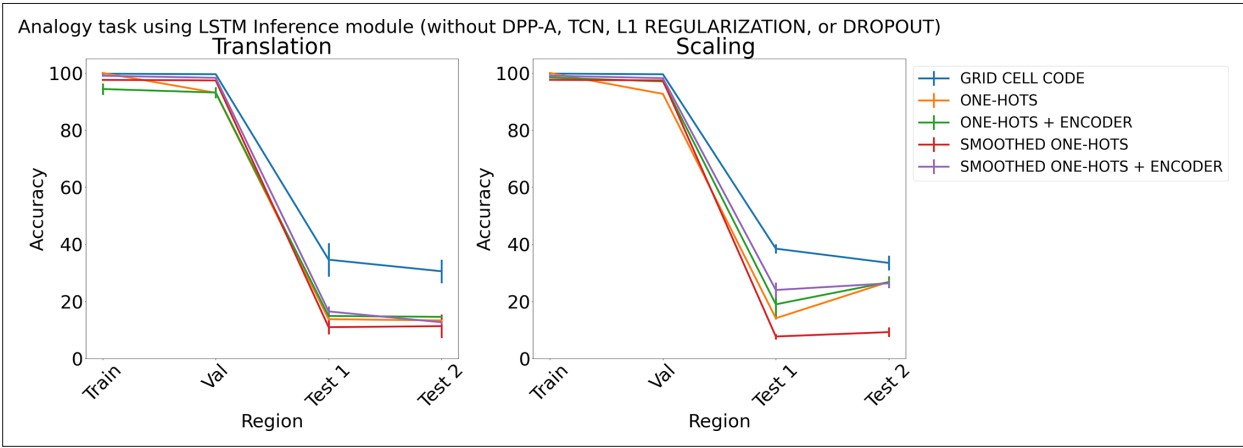

**Figure 8.** Results on analogy on each region using different embeddings (grid cell code, and one-hots or smoothed one-hots with and without an encoder) and an LSTM in the inference module, but without determinantal point process attention (DPP-A), temporal context normalization (TCN), L1 Regularization, or Dropout for translation (left) and scaling (right).

higher performance than chance (except smoothed one-hots), they still achieve lower test accuracy than grid cell code. More ablation studies can be found in Appendix 1: 'Ablation study on choice of frequency', 'Ablation on DPP-A', and Figure A5.

## Discussion

We have identified how particular properties of processing observed in the brain can be used to achieve strong OOD generalization, and introduced a two-component algorithm to promote OOD generalization in deep neural networks. The first component is a structured representation of the training data, modeled closely on known properties of grid cells – a population of cells that collectively represent abstract position using a periodic code. However, despite their intrinsic structure, we find that grid cell code and standard error-driven learning alone are insufficient to promote OOD generalization, and standard approaches for preventing overfitting offer only modest gains. This is addressed by the second component, using DPP-A to implement attentional regularization over the grid cell code. DPP-A allows only a relevant and diverse subset of cells to influence downstream computation in the inference module using the statistics of the training data. For proof of concept, we started with two challenging cognitive tasks (analogy and arithmetic), and showed that the combination of grid cell code and DPP-A promotes OOD generalization across both translation and scale when incorporated into common architectures (LSTM and transformer). It is worth noting that the DPP-A attentional mechanism is different from the attentional mechanism in the transformer module, and both are needed for strong OOD generalization in the case of transformers. Use of the standard self-attention mechanism in transformers over the inputs (i.e., A, B, C, and D for the analogy task) in place of DPP-A would lead to weightings of grid cell embeddings over all frequencies and phases. The objective function for the DPP-A represents an inductive bias, that selectively assigns the greatest weight to all grid cell embeddings (i.e., for all phases) of the frequency for which the determinant of the covariance matrix is greatest computed over the training space. The transformer inference module then attends over the inputs with the selected grid cell embeddings based on the DPP-A objective.

The current approach has some limitations and presents interesting directions for future work. First, we derive the grid cell code from known properties of neural systems, rather than obtaining the code directly from real-world data. Here, we are encouraged by the body of work providing evidence for grid cell code in the hidden layers of neural networks in a variety of task contexts and architectures (*Wei et al., 2015*; *Cueva and Wei, 2018*; *Banino et al., 2018*; *Whittington et al., 2020*). This suggests reason for optimism that DPP-A may promote strong generalization in cases where grid cell code naturally emerge: for example, navigation tasks (*Banino et al., 2018*) and reasoning by transitive inference (*Whittington et al., 2020*). Integrating our approach with structured representations acquired from high-dimensional, naturalistic datasets remains a critical next step which would have significant potential for broader future practical applications. So too does application to more complex transformations beyond translation and scale, such as rotation, and complex forms of representations, and analogical reasoning tasks (*Holyoak, 2012*; *Webb et al., 2023*; *Lu et al., 2022*). Second, it is not clear how DPP-A might be implemented in a neural network. In that regard, *Bozkurt et al., 2022* recently proposed a biologically plausible neural network algorithm using a weighted similarity matrix approach to implement a determinant maximization criterion, which is the core idea underlying the objective function we use for DPP-A (*Equation 6*), suggesting that the DPP-A mechanism we describe may also be biologically plausible. This could be tested experimentally by exposing individuals (e.g., rodents or humans) to a task that requires consistent exposure to a subregion, and evaluating the distribution of activity over the grid cells. Our model predicts that high-frequency grid cells should increase their firing rate more than low-frequency cells, since the high-frequency grid cells maximize the determinant of the covariance matrix of the grid cell embeddings. It is also worth noting that *Frankland and Cohen, 2020* have suggested that the use of DPPs may also help explain a mutual exclusivity bias observed in human word learning and reasoning. While this is not direct evidence of biological plausibility, it is consistent with the idea that the human brain selects representations for processing that maximize the volume of the representational space, which can be achieved by maximizing the DPP-A objective function defined in *Equation 6*. Third, we compared grid cell code to only one-hots and place cell code. Future work could compare to a broader range of potential biological coding schemes for the overall space, for example boundary vector cell coding (*Barry et al., 2006*), band cell coding, or egocentric boundary cell coding (*Hinman et al., 2019*).

Finally, we focus on analogies in linear spaces which limits the generality of our approach in nonlinear spaces. In that regard, there are at least two potential directions that could be pursued. One is to directly encode nonlinear structures (such as trees and rings) with grid cells, to which DPP-A could be applied as described in our model. The Tolman-Eichenbaum Machine (TEM) model (*Whittington et al., 2020*) suggests that grid cells in the medial entorhinal may form a basis set that captures structural knowledge for such nonlinear spaces, such as social hierarchies and transitive inference when formalized as a connected graph. Another would be to use eigen decomposition of the successor representation (*Dayan, 1993*), a learnable predictive representation of possible future states that has been shown by *Stachenfeld et al., 2017* to provide an abstract structured representation of a space that is analogous to the grid cell code. This general-purpose mechanism could be applied to represent analogies in nonlinear spaces (*Frankland et al., 2019*), for which there may not be a clear factorization in terms of grid cells (i.e., distinct frequencies and multiple phases within each frequency). Since the DPP-A mechanism, as we have described it, requires representations to be factored in this way it would need to be modified for such purpose. Either of these approaches, if successful, would allow our model to be extended to domains containing nonlinear forms of structure. To the extent that different coding schemes (i.e., basis sets) are needed for different forms of structure, the question of how these are identified and engaged for use in a given setting is clearly an important one, that is not addressed by the current work. We imagine that this is likely subserved by monitoring and selection mechanisms proposed to underlie the capacity for selective attention and cognitive control (*Shenhav et al., 2013*), though the specific computational mechanisms that underlie this function remain an important direction for future research.

## Additional information

### Funding

| Funder | Grant reference number | Author |
| --- | --- | --- |
| Office of Naval Research | | Shanka Subhra Mondal<br>Steven Frankland<br>Taylor W Webb<br>Jonathan D Cohen |

The funders had no role in study design, data collection, and interpretation, or the decision to submit the work for publication.

### Author contributions

Shanka Subhra Mondal, Data curation, Formal analysis, Visualization, Methodology, Writing - original draft; Steven Frankland, Taylor W Webb, Conceptualization, Supervision, Writing – review and editing; Jonathan D Cohen, Conceptualization, Supervision, Funding acquisition, Project administration, Writing – review and editing

### Author ORCIDs

Shanka Subhra Mondal ⓘ https://orcid.org/0000-0003-1252-0129
Jonathan D Cohen ⓘ https://orcid.org/0000-0003-2316-0763

Reviewer #1 (Public review): https://doi.org/10.7554/eLife.89911.3.sa1
Author response https://doi.org/10.7554/eLife.89911.3.sa2

## Additional files

### Supplementary files

• MDAR checklist

• Source data 1. The zip file contains the data for the arithmetic task.

• Source code 1. The zip file contains the code downloaded from the github repo provided earlier.

## Data availability

All data generated during this study are uploaded as source data files. Modeling code is uploaded as *Source code 1*.

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

# Appendix 1

## Code and data availability
The code and the data can be downloaded from https://github.com/Shanka123/DPP-Attention_ Grid-Cell-Code (copy archived at *Mondal, 2024*).

## More experimental details
The size of the training region was 100. For the analogy task, we used 653,216 training samples, 163,304 validation samples, and 20,000 testing samples for each of the nine regions. For the arithmetic task, we used 80,000 training samples, 20,000 validation samples, and 20,000 testing samples for each of the nine regions with equal number of addition and multiplication problems. We used the PyTorch library (*Paszke et al., 2017*) for all experiments. For each network, the training epoch that achieved the best validation accuracy was used to report performance accuracy for the training stimulus sets, validation sets (held out stimuli from the training range), and OOD generalization test sets (from regions beyond the range of the training data).

## Why is OOD generalization performance worse for the multiplication task?
In an effort to understand why DPP-A achieved around 65% average test accuracy on multiplication compared to nearly perfect accuracy for addition and analogy tasks, we analyzed the distribution of the grid cell embeddings for the frequency which had the maximum within-frequency determinant at the end of the first step in Algorithm 1. More specifically for $A$, $B$, and the correct answer $C$, we analyzed the distribution of each grid cell for the training and the nine test regions. Note that since the total number of grid cells was 900 and there were nine frequencies, the dimension of the grid cell embeddings corresponding to $f_{max_{DPP}}$ grid cell frequency was 100. To quantify the similarity between training and the test distributions, we computed cosine distance (1 − cosine similarity), and averaged it over the 100 dimensions and nine test regions. We found that the average cosine distance is 5× greater for multiplication than addition problem (0.0002 for addition: 0.001 for multiplication). In this respect, grid cell code does not perfectly preserve the relational structure of the multiplication problem, which we would expect to limit DPP-A's OOD generalization ability in that task domain.

## Ablation study on choice of frequency

Analogy task using LSTM Inference module (with DPP-A selecting top $K$ frequencies)

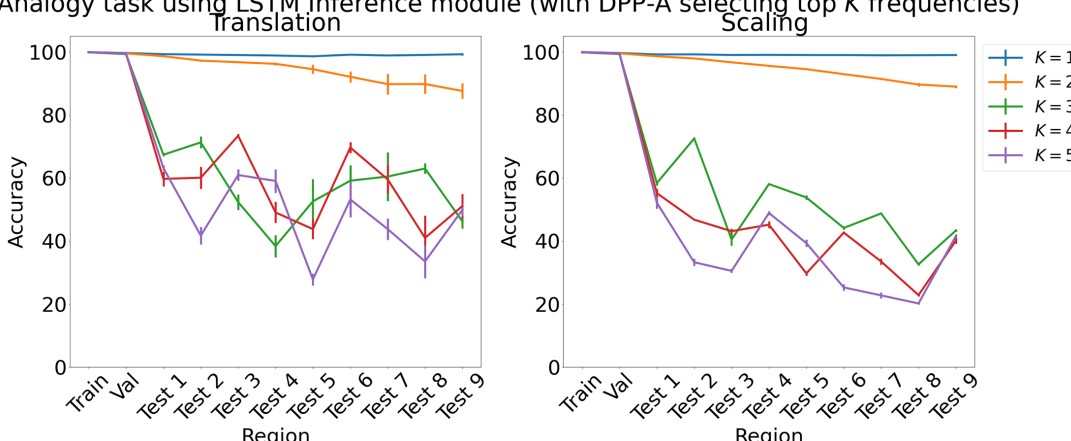

**Appendix 1—figure 1.** Results on analogy on each region using LSTM in the inference module for choosing top $K$ frequencies with $\hat{F}_f$ in Algorithm 1. Results show mean accuracy on each region averaged over three trained networks along with errorbar (standard error of the mean).

## Baseline using dynamic attention across frequencies

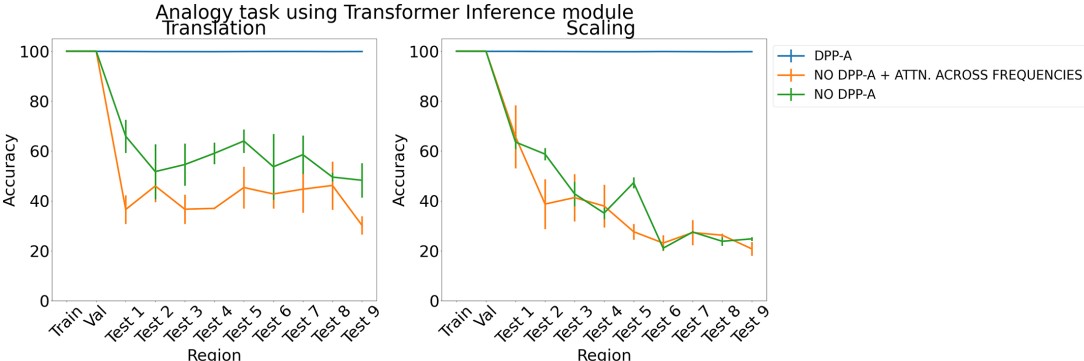

**Appendix 1—figure 2.** Results on analogy on each region for translation and scaling using the transformer in the inference module.

## Ablation study on arithmetic task

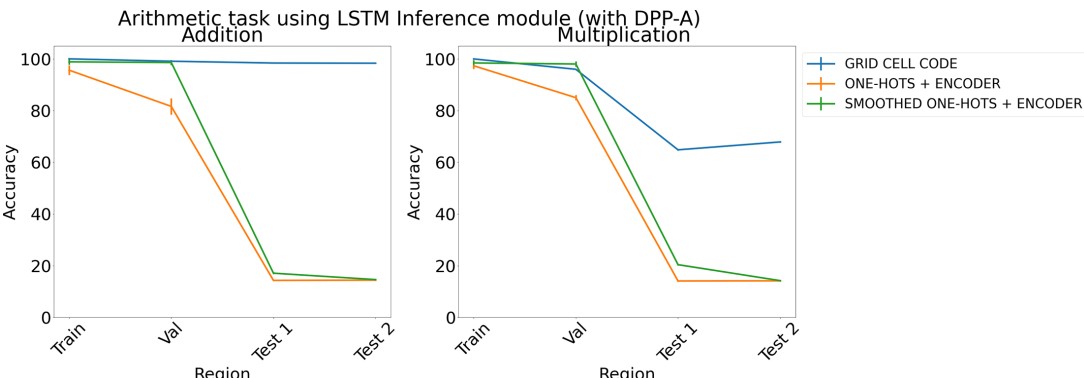

**Appendix 1—figure 3.** Results on arithmetic with different embeddings (with determinantal point process attention [DPP-A]) using LSTM in the inference module. Results show mean accuracy on each region averaged over three trained networks along with errorbar (standard error of the mean).

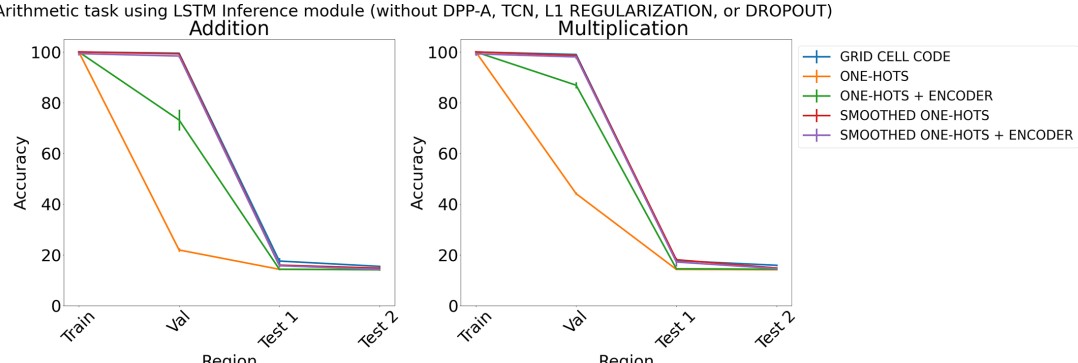

**Appendix 1—figure 4.** Results on arithmetic with different embeddings (without determinantal point process attention [DPP-A], temporal context normalization [TCN], L1 Regularization, or Dropout) using LSTM in the inference module. Results show mean accuracy on each region averaged over three trained networks along with errorbar (standard error of the mean).

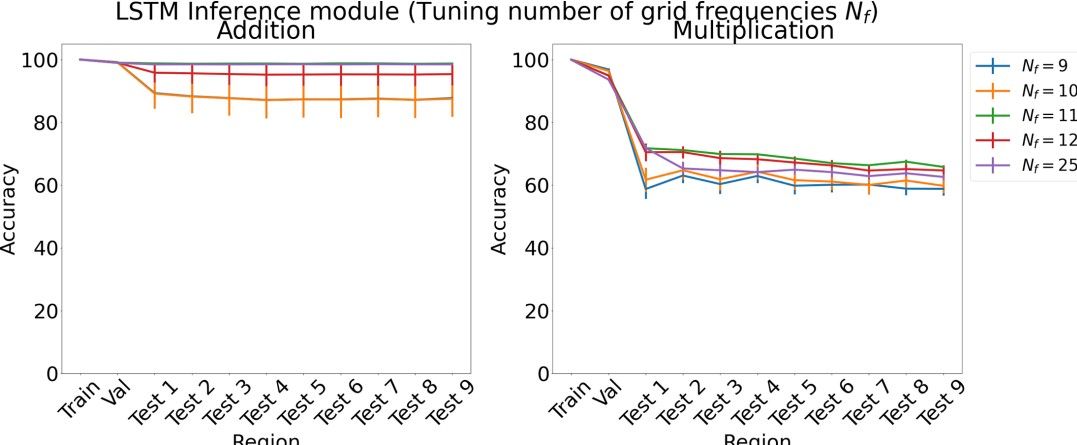

**Appendix 1—figure 5.** Results on arithmetic for increasing number of grid cell frequencies $N_f$ on each region using LSTM in the inference module. Results show mean accuracy on each region averaged over three trained networks along with errorbar (standard error of the mean).

## Regression formulation

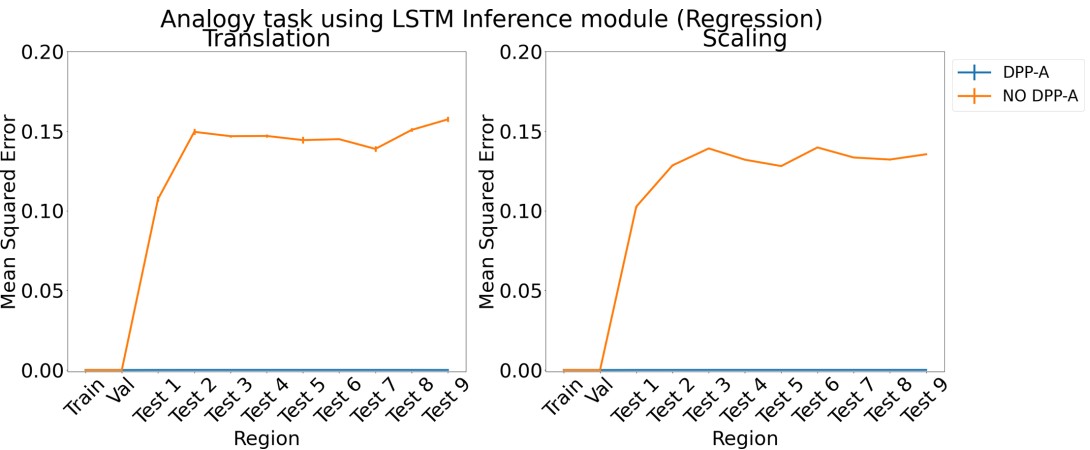

**Appendix 1—figure 6.** Results for regression on analogy using LSTM in the inference module. Results show mean squared error on each region averaged over three trained networks along with errorbar (standard error of the mean).

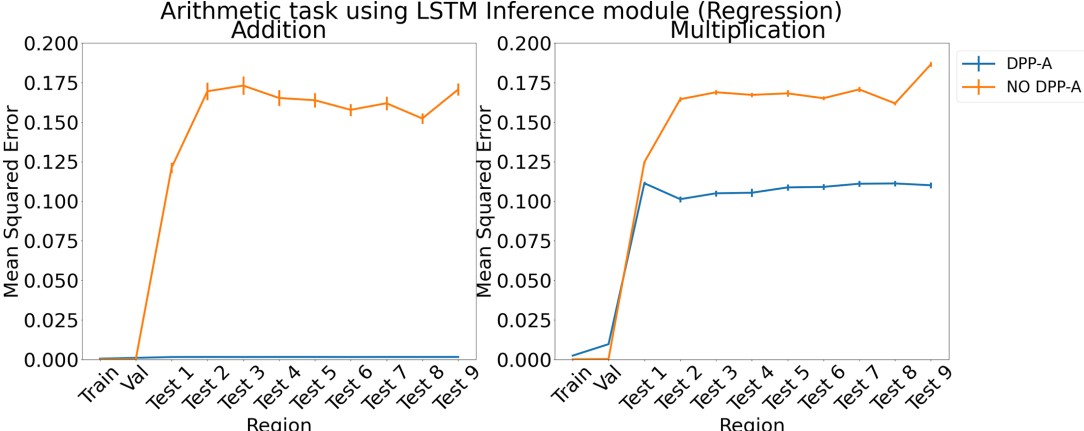

**Appendix 1—figure 7.** Results for regression on arithmetic on each region using LSTM in the inference module. Results show mean squared error on each region averaged over three trained networks along with errorbar (standard error of the mean).

We also tried formulating the analogy and arithmetic tasks as regression instead of classification via a scoring mechanism. For DPP-A, the inference module was trained to generate the grid cell embeddings belonging to the $f_{max_{DPP}}$ frequency, which had the maximum within-frequency determinant at the end of the first step in Algorithm 1 for the correct completion, given as input the $f_{max_{DPP}}$ frequency grid cell embeddings for $A$, $B$, $C$ for the analogy task and $A$, $B$ for the arithmetic task. A linear layer with 100 units and sigmoid activation was used to generate the output of the inference module and was trained to minimize the mean squared error with the $f_{max_{DPP}}$ frequency grid cell embeddings of the correct completion. We compared DPP-A with a version that didn't use the attentional objective (no DPP-A), where the inference module was trained to generate the grid cell embeddings for all the frequencies, but was evaluated on only the $f_{max_{DPP}}$ frequency grid cell embeddings for fair comparison with the DPP-A version. *Appendix 1—figure 6* shows the results for the analogy task using an LSTM in the inference module. For both the translation (left) and scaling (right) regimes, DPP-A achieves nearly zero mean squared error on all the test regions, considerably outperforming the no DPP-A which achieves a much higher error. *Appendix 1—figure 7* shows the results for arithmetic problems using an LSTM in the inference module. For addition problems, shown on the left, DPP-A achieves nearly zero mean squared error on the test regions. For multiplication problems, shown on the right, DPP-A achieves a lower mean squared error on the test regions, 0.11, compared to no DPP-A which achieves around 0.17.

## Effect of L1 regularization strength ($\lambda$)

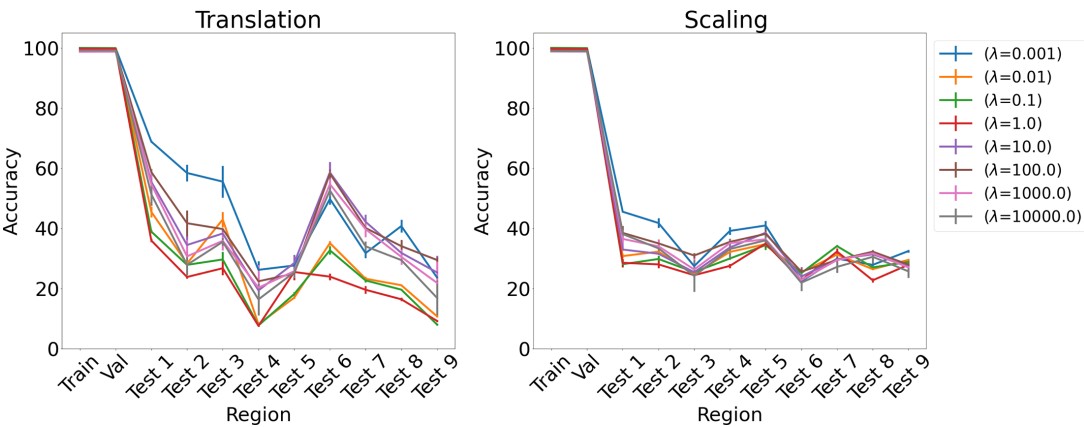

**Appendix 1—figure 8.** Results on analogy for L1 regularization for various $\lambda$s for translation and scaling using LSTM in the inference module. Results show mean accuracy on each region averaged over three trained networks along with errorbar (standard error of the mean).

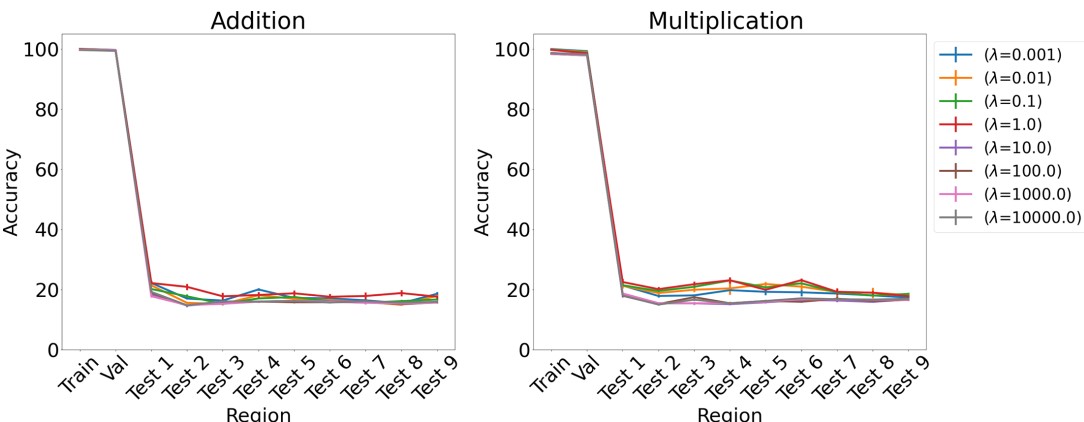

**Appendix 1—figure 9.** Results on arithmetic for L1 regularization for various $\lambda$s using LSTM in the inference module. Results show mean accuracy on each region averaged over three trained networks along with errorbar (standard error of the mean).

## Ablation on DPP-A

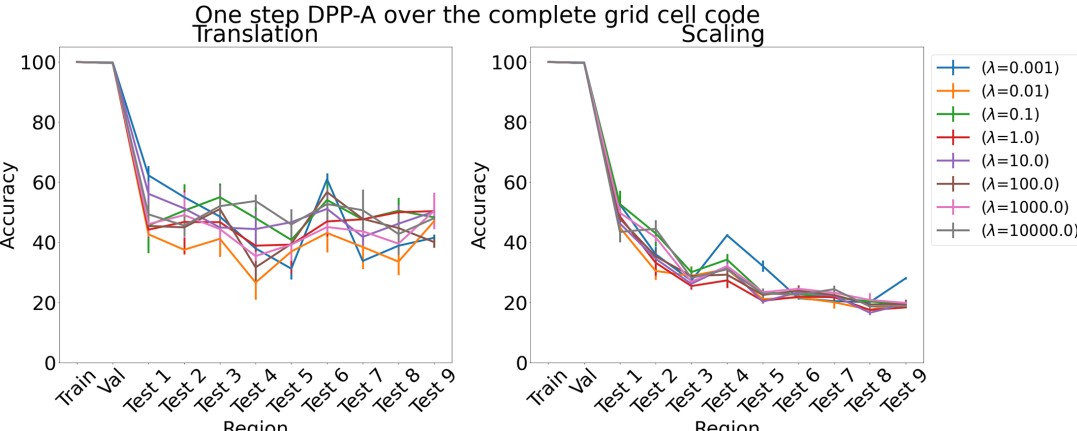

**Appendix 1—figure 10.** Results on analogy for one step determinantal point process attention (DPP-A) over the complete grid cell code for various $\lambda$s for translation and scaling using LSTM in the inference module. Results show mean accuracy on each region averaged over three trained networks along with errorbar (standard error of the mean).

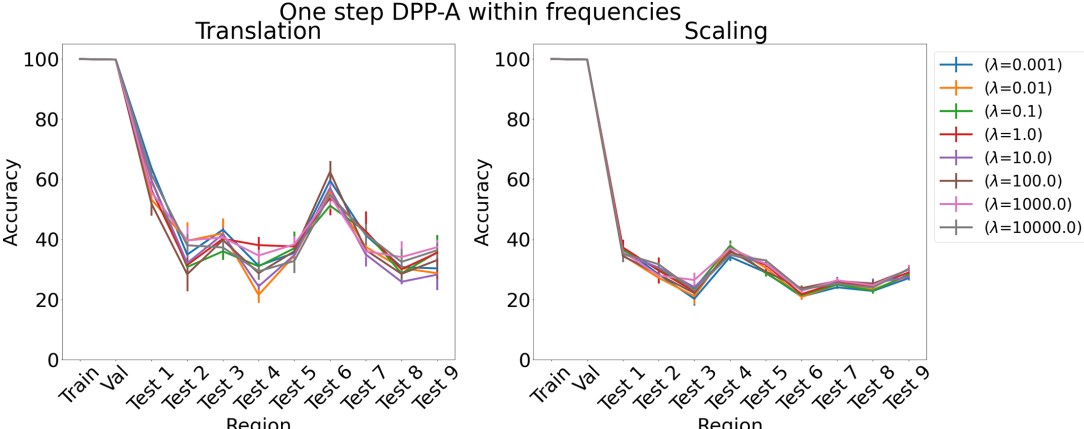

**Appendix 1—figure 11.** Results on analogy for one step determinantal point process attention (DPP-A) within frequencies for various λs for translation and scaling using LSTM in the inference module. Results show mean accuracy on each region averaged over three trained networks along with errorbar (standard error of the mean).

The proposed DPP-A method (Algorithm 1) consists of two steps with $\mathcal{L}_{DPP}$ in the first step and $\mathcal{L}_{task}$ in the second step. We considered two ablation experiments which consists of a single step. In one case, we maximized the objective function, $\hat{F}(\boldsymbol{g}, \boldsymbol{V}) = \log \det(\mathrm{diag}(\sigma(\mathbf{g}))(\boldsymbol{V} - \boldsymbol{I}) + \boldsymbol{I})$, over the grid cell embeddings of all frequencies and phases (instead of summing $\hat{F}$ corresponding to the grid cell embeddings from each frequency independently as done in the first step of Algorithm 1), using stochastic gradient ascent, along with minimizing $\mathcal{L}_{task}$, which would use all the attended grid cell embeddings (instead of using $f_{max_{DPP}}$ frequency grid cell embeddings as done in the second step of Algorithm 1). So total loss, $\mathcal{L} = \mathcal{L}_{task} - \lambda * \hat{F}(\boldsymbol{g}, \boldsymbol{V})$. We refer to this ablation experiment as one step DPP-A over the complete grid cell code. The results on the analogy for this ablation experiment are shown in *Appendix 1—figure 10*. We see that the accuracy on test analogies for translation for various λs is around 30–60%, and for scaling around 20–40%, which is much lower than the nearly perfect accuracy achieved by the proposed DPP-A method. In the other case, we maximized the objective function $\hat{F}(\boldsymbol{g}, \boldsymbol{V}, N_f, N_p) = \sum_{f=1}^{N_f} \log \det(\mathrm{diag}(\sigma(\boldsymbol{g_f}))(\boldsymbol{V_f} - \boldsymbol{I}) + \boldsymbol{I})$, using stochastic gradient ascent, which is same as $\mathcal{L}_{DPP}$ in the first step of Algorithm 1, along with minimizing $\mathcal{L}_{task}$, which would use all the attended grid cell embeddings. So total loss, $\mathcal{L} = \mathcal{L}_{task} - \lambda * \hat{F}(\boldsymbol{g}, \boldsymbol{V}, N_f, N_p)$. We refer to this ablation experiment as one step DPP-A within frequencies. As shown in *Appendix 1—figure 11*, the accuracy on test analogies for both translation and scaling for various λs is in a similar range to one step DPP-A over the complete grid cell code, and is much lower than the nearly perfect accuracy achieved by the proposed DPP-A method.

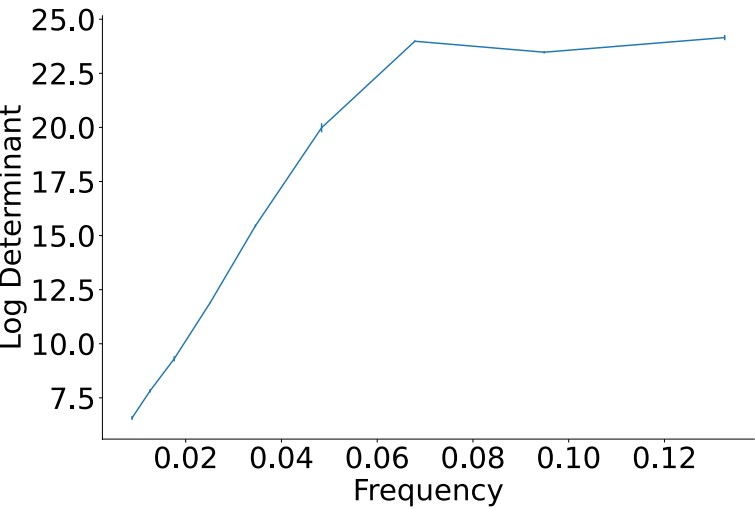

**Appendix 1—figure 12.** Approximate maximum log determinant of the covariance matrix over the grid cell embeddings (*y*-axis) for each frequency (*x*-axis), obtained after maximizing *Equation 6*.

The gates within each frequency were optimized (independent of the task inputs), according to *Equation 6*, to compute the approximate maximum within-frequency log determinant of the covariance matrix over the grid cell embeddings individually for each frequency. We then used the grid cell embeddings belonging to the frequency that had the maximum within-frequency log determinant for training the inference module, which always happened to be grid cells within the top three frequencies. *Appendix 1—figure 12* shows the approximate maximum log determinant (on the y-axis) for the different frequencies (on the x-axis). The intuition behind why DPP-A identified grid cell embeddings corresponding to the highest spatial frequencies, and why this produced the best OOD generalization (i.e., extrapolation on our analogy tasks), is because those grid cell embeddings exhibited greater variance over the training data than the lower-frequency embeddings, while at the same time the correlations among those grid cell embeddings were lower than the correlations among the lower-frequency grid cell embeddings. The determinant of the covariance matrix of the grid cell embeddings is maximized when the variances of the grid cell embeddings are high (they are 'expressive') and the correlation among the grid cell embeddings is low (they 'cover the representational space'). As a result, the higher-frequency grid cell embeddings more efficiently covered the representational space of the training data, allowing them to efficiently capture the same relational structure across training and test distributions which is required for OOD generalization.

To demonstrate that the higher grid cell frequencies more efficiently cover the representational space, we analyzed in *Appendix 1—figure 13*, the results after the summation of the multiplication of the grid cell embeddings over the 2D space of 1000 × 1000 locations, with their corresponding gates for three representative frequencies (left, middle, and right panels showing results for the lowest, middle, and highest grid cell frequencies, respectively, of the nine used in the model), obtained after maximizing *Equation 6* for each grid cell frequency. The color code indicates the responsiveness of the grid cells to different X and Y locations in the input space (lighter color corresponding to greater responsiveness). Note that the dark blue area (denoting regions of least responsiveness to any grid cell) is greatest for the lowest frequency and nearly zero for the highest frequency, illustrating that grid cell embeddings belonging to the highest frequency more efficiently cover the representational space which allows them to capture the same relational structure across training and test distributions as required for OOD generalization.

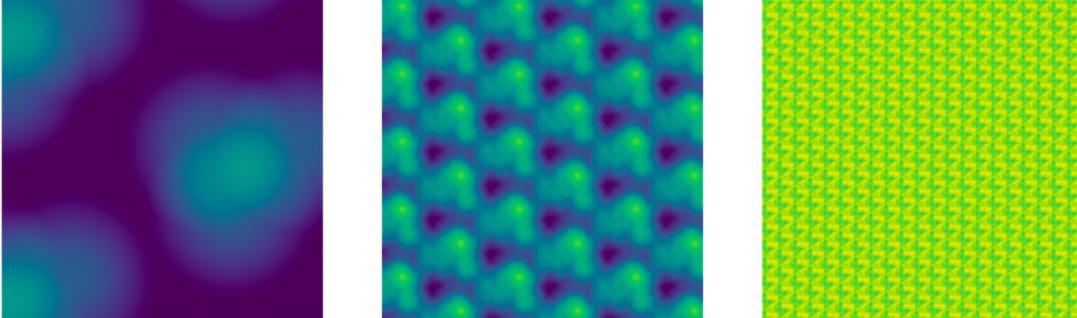

**Appendix 1—figure 13.** Each panel shows the results after summation of the multiplication of the grid cell embeddings over the 2D space of 1000 × 1000 locations, with their corresponding gates for a particular frequency, obtained after maximizing *Equation 6* for each grid cell frequency. The left, middle, and right panels show results for the lowest, middle, and highest grid cell frequencies, respectively, of the nine used in the model. Lighter color in each panel corresponds to greater responsiveness of grid cells at that particular location in the 2D space.

