## [Editor Report · eLife assessment]

This **important** modeling work demonstrates out-of-distribution generalization using a grid cell coding scheme combined with an attentional mechanism that operates over these representations (Determinantal Point Process Attention). The simulations provide **compelling** evidence that the model can improve generalization performance for analogies, addition, and multiplication. The paper is significant in demonstrating how neural grid codes can support human-like generalization capabilities in analogy and arithmetic tasks, which has been a challenge for prior models.

---

## [Referee Report · Reviewer #1 (Public review)]

This paper presents a cognitive model of out-of-distribution generalisation, where the representational basis is grid-cell codes. In particular, the authors consider the tasks of analogies, addition, and multiplication, and the out-of-distribution tests are shifting or scaling the input domain. The authors utilise grid cell codes, which are multi-scale as well as translationally invariant due to their periodicity. To allow for domain adaptation, the authors use DPP-A which is, in this context, a mechanism of adapting to input scale changes. The authors present simulations results demonstrating this model can perform out-of-distribution generalisation to input translations and re-scaling, whereas other models fail.

This paper makes the point it sets out to - that there are some underlying representational bases, like grid cells, that when combined with a domain adaptation mechanism, like DPP-A, can facilitate out-of-generalisation. I don't have any issues with the technical details.

The paper nicely demonstrates how neural codes can be transformed into a common representational space so that analogies, and presumably other useful tasks/computations, can be performed.

---

## [Author Response]

The following is the authors’ response to the original reviews.

**Major comments (Public Reviews)**

**Generality of grid cells**

We appreciate the reviewers’ concern regarding the generality of our approach, and in particular for analogies in nonlinear spaces. In that regard, there are at least two potential directions that could be pursued. One is to directly encode nonlinear structures (such as trees, rings, etc.) with grid cells, to which DPP-A could be applied as described in our model. The TEM model [1] suggests that grid cells in the medial entorhinal may form a basis set that captures structural knowledge for such nonlinear spaces, such as social hierarchies and transitive inference when formalized as a connected graph. Another would be to use eigen-decomposition of the successor representation [2], a learnable predictive representation of possible future states that has been shown by Stachenfield et al. [3] to provide an abstract structured representation of a space that is analogous to the grid cell code. This general-purpose mechanism could be applied to represent analogies in nonlinear spaces [4], for which there may not be a clear factorization in terms of grid cells (i.e., distinct frequencies and multiple phases within each frequency). Since the DPP-A mechanism, as we have described it, requires representations to be factored in this way it would need to be modified for such purpose. Either of these approaches, if successful, would allow our model to be extended to domains containing nonlinear forms of structure. To the extent that different coding schemes (i.e., basis sets) are needed for different forms of structure, the question of how these are identified and engaged for use in a given setting is clearly an important one, that is not addressed by the current work. We imagine that this is likely subserved by monitoring and selection mechanisms proposed to underlie the capacity for selective attention and cognitive control [5], though the specific computational mechanisms that underlie this function remain an important direction for future research. We have added a discussion of these issues in Section 6 of the updated manuscript.

(1) Whittington, J.C., Muller, T.H., Mark, S., Chen, G., Barry, C., Burgess, N. and Behrens, T.E., 2020. The Tolman-Eichenbaum machine: unifying space and relational memory through generalization in the hippocampal formation. Cell, 183(5), pp.1249-1263.

(2) Dayan, P., 1993. Improving generalization for temporal difference learning: The successor representation. Neural computation, 5(4), pp.613-624.

(3) Stachenfeld, K.L., Botvinick, M.M. and Gershman, S.J., 2017. The hippocampus as a predictive map. Nature neuroscience, 20(11), pp.1643-1653.

(4) Frankland, S., Webb, T.W., Petrov, A.A., O'Reilly, R.C. and Cohen, J., 2019. Extracting and Utilizing Abstract, Structured Representations for Analogy. In CogSci (pp. 1766-1772).

(5) Shenhav, A., Botvinick, M.M. and Cohen, J.D., 2013. The expected value of control: an integrative theory of anterior cingulate cortex function. Neuron, 79(2), pp.217-240.

**Biological plausibility of DPP-A**

We appreciate the reviewers’ interest in the biological plausibility of our model, and in particular the question of whether and how DPP-A might be implemented in a neural network. In that regard, Bozkurt et al. [1] recently proposed a biologically plausible neural network algorithm using a weighted similarity matrix approach to implement a determinant maximization criterion, which is the core idea underlying the objective function we use for DPP-A, suggesting that the DPP-A mechanism we describe may also be biologically plausible. This could be tested experimentally by exposing individuals (e.g., rodents or humans) to a task that requires consistent exposure to a subregion, and evaluating the distribution of activity over the grid cells. Our model predicts that high frequency grid cells should increase their firing rate more than low frequency cells, since the high frequency grid cells maximize the determinant of the covariance matrix of the grid cell embeddings. It is also worth noting that Frankland et al. [2] have suggested that the use of DPPs may also help explain a mutual exclusivity bias observed in human word learning and reasoning. While this is not direct evidence of biological plausibility, it is consistent with the idea that the human brain selects representations for processing that maximize the volume of the representational space, which can be achieved by maximizing the DPP-A objective function defined in Equation 6. We have added a comment to this effect in Section 6 of the updated manuscript.

(1) Bozkurt, B., Pehlevan, C. and Erdogan, A., 2022. Biologically-plausible determinant maximization neural networks for blind separation of correlated sources. Advances in Neural Information Processing Systems, 35, pp.13704-13717.

(2) Frankland, S. and Cohen, J., 2020. Determinantal Point Processes for Memory and Structured Inference. In CogSci.

**Simplicity of analogical problem and comparison to other models using this task**

First, we would like to point out that analogical reasoning is a signatory feature of human cognition, which supports flexible and efficient adaptation to novel inputs that remains a challenge for most current neural network architectures. While humans can exhibit complex and sophisticated forms of analogical reasoning [1, 2, 3], here we focused on a relatively simple form, that was inspired by Rumelhart’s parallelogram model of analogy [4,5] that has been used to explain traditional human verbal analogies (e.g., “king is to what as man is to woman?”). Our model, like that one, seeks to explain analogical reasoning in terms of the computation of simple Euclidean distances (i.e., A - B = C - D, where A, B, C, D are vectors in 2D space). We have now noted this in Section 2.1.1 of the updated manuscript. It is worth noting that, despite the seeming simplicity of this construction, we show that standard neural network architectures (e.g., LSTMs and transformers) struggle to generalize on such tasks without the use of the DPP-A mechanism.

Second, we are not aware of any previous work other than Frankland et al. [6] cited in the first paragraph of Section 2.2.1, that has examined the capacity of neural network architectures to perform even this simple form of analogy. The models in that study were hardcoded to perform analogical reasoning, whereas we trained models to learn to perform analogies. That said, clearly a useful line of future work would be to scale our model further to deal with more complex forms of representation and analogical reasoning tasks [1,2,3]. We have noted this in Section 6 of the updated manuscript.

(1) Holyoak, K.J., 2012. Analogy and relational reasoning. The Oxford handbook of thinking and reasoning, pp.234-259.

(2) Webb, T., Fu, S., Bihl, T., Holyoak, K.J. and Lu, H., 2023. Zero-shot visual reasoning through probabilistic analogical mapping. Nature Communications, 14(1), p.5144.

(3) Lu, H., Ichien, N. and Holyoak, K.J., 2022. Probabilistic analogical mapping with semantic relation networks. Psychological review.

(4) Rumelhart, D.E. and Abrahamson, A.A., 1973. A model for analogical reasoning. Cognitive Psychology, 5(1), pp.1-28.

(5) Mikolov, T., Chen, K., Corrado, G. and Dean, J., 2013. Efficient estimation of word representations in vector space. arXiv preprint arXiv:1301.3781.

(6) Frankland, S., Webb, T.W., Petrov, A.A., O'Reilly, R.C. and Cohen, J., 2019. Extracting and Utilizing Abstract, Structured Representations for Analogy. In CogSci (pp. 1766-1772).

**Clarification of DPP-A attentional modulation**

We would like to clarify several concerns regarding the DPP-A attentional modulation. First, we would like to make it clear that ω is not meant to correspond to synaptic weights, and thank the reviewer for noting the possibility for confusion on this point. It is also distinct from a biasing input, which is often added to the product of the input features and weights. Rather, in our model ω is a vector, and diag (ω) converts it into a matrix with ω as the diagonal of the matrix, and the rest entries are zero. In Equation 6, diag(ω) is matrix multiplied with the covariance matrix V, which results in elementwise multiplication of ω with column vectors of V, and hence acts more like gates. We have noted this in Section 2.2.2 and have changed all instances of “weights (ω)” to “gates (ɡ)” in the updated manuscript. We have also rewritten the definition of Equation 6 and uses of it (as in Algorithm 1) to depict the use of sigmoid nonlinearity (σ) to , so that the resulting values are always between 0 and 1.

Second, we would like to clarify that we don’t compute the inner product between the gates ɡ and the grid cell embeddings x anywhere in our model. The gates within each frequency were optimized (independent of the task inputs), according to Equation 6, to compute the approximate maximum log determinant of the covariance matrix over the grid cell embeddings individually for each frequency. We then used the grid cell embeddings belonging to the frequency that had the maximum within-frequency log determinant for training the inference module, which always happened to be grid cells within the top three frequencies. Author response image 1 (also added to the Appendix, Section 7.10 of the updated manuscript) shows the approximate maximum log determinant (on the y-axis) for the different frequencies (on the x-axis).

**Author response image 1. sa2fig1:** Approximate maximum log determinant of the covariance matrix over the grid cell embeddings (y-axis) for each frequency (x-axis), obtained after maximizing Equation 6.

Third, we would like to clarify our interpretation of why DPP-A identified grid cell embeddings corresponding to the highest spatial frequencies, and why this produced the best OOD generalization (i.e., extrapolation on our analogy tasks). It is because those grid cell embeddings exhibited greater variance over the training data than the lower frequency embeddings, while at the same time the correlations among those grid cell embeddings were lower than the correlations among the lower frequency grid cell embeddings. The determinant of the covariance matrix of the grid cell embeddings is maximized when the variances of the grid cell embeddings are high (they are “expressive”) and the correlation among the grid cell embeddings is low (they “cover the representational space”). As a result, the higher frequency grid cell embeddings more efficiently covered the representational space of the training data, allowing them to efficiently capture the same relational structure across training and test distributions which is required for OOD generalization. We have added some clarification to the second paragraph of Section 2.2.2 in the updated manuscript. Furthermore, to illustrate this graphically, Author response image 2 (added to the Appendix, Section 7.10 of the updated manuscript) shows the results after the summation of the multiplication of the grid cell embeddings over the 2d space of 1000x1000 locations, with their corresponding gates for 3 representative frequencies (left, middle and right panels showing results for the lowest, middle and highest grid cell frequencies, respectively, of the 9 used in the model), obtained after maximizing Equation 6 for each grid cell frequency. The color code indicates the responsiveness of the grid cells to different X and Y locations in the input space (lighter color corresponding to greater responsiveness). Note that the dark blue area (denoting regions of least responsiveness to any grid cell) is greatest for the lowest frequency and nearly zero for the highest frequency, illustrating that grid cell embeddings belonging to the highest frequency more efficiently cover the representational space which allows them to capture the same relational structure across training and test distributions as required for OOD generalization.

**Author response image 2. sa2fig2:** Each panel shows the results after summation of the multiplication of the grid cell embeddings over the 2d space of 1000x1000 locations, with their corresponding gates for a particular frequency, obtained after maximizing Equation 6 for each grid cell frequency. The left, middle, and right panels show results for the lowest, middle, and highest grid cell frequencies, respectively, of the 9 used in the model. Lighter color in each panel corresponds to greater responsiveness of grid cells at that particular location in the 2d space.

Finally, we would like to clarify how the DPP-A attentional mechanism is different from the attentional mechanism in the transformer module, and why both are needed for strong OOD generalization. Use of the standard self-attention mechanism in transformers over the inputs (i.e., A, B, C, and D for the analogy task) in place of DPP-A would lead to weightings of grid cell embeddings over all frequencies and phases. The objective function for the DPP-A represents an inductive bias, that selectively assigns the greatest weight to all grid cell embeddings (i.e., for all phases) of the frequency for which the determinant of the covariance matrix is greatest computed over the training space. The transformer inference module then attends over the inputs with the selected grid cell embeddings based on the DPP-A objective. We have added a discussion of this point in Section 6 of the updated manuscript.

We would like to thank the reviewers for their recommendations. We have tried our best to incorporate them into our updated manuscript. Below we provide a detailed response to each of the recommendations grouped for each reviewer.

**Reviewer #1 (Recommendations for the authors)**
(1) It would be helpful to see some equations for R in the main text.

We thank the reviewer for this suggestion. We have now added some equations explaining the working of R in Section 2.2.3 of the updated manuscript.

(2) Typo: p 11 'alongwith' -> 'along with'

We have changed all instances of ‘alongwith’ to ‘along with’ in the updated manuscript.

(3) Presumably, this is related to equivariant ML - it would be helpful to comment on this.

Yes, this is related to equivariant ML, since the properties of equivariance hold for our model. Specifically, the probability distribution after applying softmax remains the same when the transformation (translation or scaling) is applied to the scores for each of the answer choices obtained from the output of the inference module, and when the same transformation is applied to the stimuli for the task and all the answer choices before presenting as input to the inference module to obtain the scores. We have commented on this in Section 2.2.3 of the updated manuscript.

**Reviewer #2 (Recommendations for the authors)**
(1) Page 2 - "Webb et al." temporal context - they should also cite and compare this to work by Marc Howard on generalization based on multi-scale temporal context.

While we appreciate the important contributions that have been made by Marc Howard and his colleagues to temporal coding and its role in episodic memory and hippocampal function, we would like to clarify that his temporal context model is unrelated to the temporal context normalization developed by Webb et al. (2020) and mentioned on Page 2. The former (Temporal Context Model) is a computational model that proposes a role for temporal coding in the functions of the medial temporal lobe in support of episodic recall, and spatial navigation. The latter (temporal context normalization) is a normalization procedure proposed for use in training a neural network, similar to batch normalization [1], in which tensor normalization is applied over the temporal instead of the batch dimension, which is shown to help with OOD generalization. We apologize for any confusion engendered by the similarity of these terms, and failure to clarify the difference between these, that we have now attempted to do in a footnote on Page 2.

Ioffe, S. and Szegedy, C., 2015, June. Batch normalization: Accelerating deep network training by reducing internal covariate shift. In International conference on machine learning (pp. 448-456). pmlr.

(2) page 3 - "known to be implemented in entorhinal" - It's odd that they seem to avoid citing the actual biology papers on grid cells. They should cite more of the grid cell recording papers when they mention the entorhinal cortex (i.e. Hafting et al., 2005; Barry et al., 2007; Stensola et al., 2012; Giocomo et al., 2011; Brandon et al., 2011).

We have now cited the references mentioned below, on page 3 after the phrase “known to be implemented in entohinal cortex”.

(1) Barry, C., Hayman, R., Burgess, N. and Jeffery, K.J., 2007. Experience-dependent rescaling of entorhinal grids. Nature neuroscience, 10(6), pp.682-684.

(2) Stensola, H., Stensola, T., Solstad, T., Frøland, K., Moser, M.B. and Moser, E.I., 2012. The entorhinal grid map is discretized. Nature, 492(7427), pp.72-78.

(3) Giocomo, L.M., Hussaini, S.A., Zheng, F., Kandel, E.R., Moser, M.B. and Moser, E.I., 2011. Grid cells use HCN1 channels for spatial scaling. Cell, 147(5), pp.1159-1170.

(4) Brandon, M.P., Bogaard, A.R., Libby, C.P., Connerney, M.A., Gupta, K. and Hasselmo, M.E., 2011. Reduction of theta rhythm dissociates grid cell spatial periodicity from directional tuning. Science, 332(6029), pp.595-599.

(3) To enhance the connection to biological systems, they should cite more of the experimental and modeling work on grid cell coding (for example on page 2 where they mention relational coding by grid cells). Currently, they tend to cite studies of grid cell relational representations that are very indirect in their relationship to grid cell recordings (i.e. indirect fMRI measures by Constaninescu et al., 2016 or the very abstract models by Whittington et al., 2020). They should cite more papers on actual neurophysiological recordings of grid cells that suggest relational/metric representations, and they should cite more of the previous modeling papers that have addressed relational representations. This could include work on using grid cell relational coding to guide spatial behavior (e.g. Erdem and Hasselmo, 2014; Bush, Barry, Manson, Burges, 2015). This could also include other papers on the grid cell code beyond the paper by Wei et al., 2015 - they could also cite work on the efficiency of coding by Sreenivasan and Fiete and by Mathis, Herz, and Stemmler.

We thank the reviewer for bringing the additional references to our attention. We have cited the references mentioned below on page 2 of the updated manuscript.

(1) Erdem, U.M. and Hasselmo, M.E., 2014. A biologically inspired hierarchical goal directed navigation model. Journal of Physiology-Paris, 108(1), pp.28-37.

(2) Sreenivasan, S. and Fiete, I., 2011. Grid cells generate an analog error-correcting code for singularly precise neural computation. Nature neuroscience, 14(10), pp.1330-1337.

(3) Mathis, A., Herz, A.V. and Stemmler, M., 2012. Optimal population codes for space: grid cells outperform place cells. Neural computation, 24(9), pp.2280-2317.

(4) Bush, D., Barry, C., Manson, D. and Burgess, N., 2015. Using grid cells for navigation. Neuron, 87(3), pp.507-520

(4) Page 3 - "Determinantal Point Processes (DPPs)" - it is rather annoying that DPP is defined after DPP-A is defined. There ought to be a spot where the definition of DPP-A is clearly stated in a single location.

We agree it makes more sense to define Determinantal Point Process (DPP) before DPP-A. We have now rephrased the sentences accordingly. In the “Abstract”, the sentence now reads “Second, we propose an attentional mechanism that operates over the grid cell code using Determinantal Point Process (DPP), which we call DPP attention (DPP-A) - a transformation that ensures maximum sparseness in the coverage of that space.” We have also modified the second paragraph of the “Introduction”. The modified portion now reads “(2) an attentional objective inspired from Determinantal Point Processes (DPPs), which are probabilistic models of repulsion arising in quantum physics [1], to attend to abstract representations that have maximum variance and minimum correlation among them, over the training data. We refer to this as DPP attention or DPP-A.” Due to this change, we removed the last sentence of the fifth paragraph of the “Introduction”.

(1) Macchi, O., 1975. The coincidence approach to stochastic point processes. Advances in Applied Probability, 7(1), pp.83-122.

(5) Page 3 - "the inference module R" - there should be some discussion about how this component using LSTM or transformers could relate to the function of actual brain regions interacting with entorhinal cortex. Or if there is no biological connection, they should state that this is not seen as a biological model and that only the grid cell code is considered biological.

While we agree that the model is not construed to be as specific about the implementation of the R module, we assume that — as a standard deep learning component — it is likely to map onto neocortical structures that interact with the entorhinal cortex and, in particular, regions of the prefrontal-posterior parietal network widely believed to be involved in abstract relational processes [1,2,3,4]. In particular, the role of the prefrontal cortex in the encoding and active maintenance of abstract information needed for task performance (such as rules and relations) has often been modeled using gated recurrent networks, such as LSTMs [5,6], and the posterior parietal cortex has long been known to support “maps” that may provide an important substrate for computing complex relations [4]. We have added some discussion about this in Section 2.2.3 of the updated manuscript.

(1) Waltz, J.A., Knowlton, B.J., Holyoak, K.J., Boone, K.B., Mishkin, F.S., de Menezes Santos, M., Thomas, C.R. and Miller, B.L., 1999. A system for relational reasoning in human prefrontal cortex. Psychological science, 10(2), pp.119-125.

(2) Christoff, K., Prabhakaran, V., Dorfman, J., Zhao, Z., Kroger, J.K., Holyoak, K.J. and Gabrieli, J.D., 2001. Rostrolateral prefrontal cortex involvement in relational integration during reasoning. Neuroimage, 14(5), pp.1136-1149.

(3) Knowlton, B.J., Morrison, R.G., Hummel, J.E. and Holyoak, K.J., 2012. A neurocomputational system for relational reasoning. Trends in cognitive sciences, 16(7), pp.373-381.

(4) Summerfield, C., Luyckx, F. and Sheahan, H., 2020. Structure learning and the posterior parietal cortex. Progress in neurobiology, 184, p.101717.

(5) Frank, M.J., Loughry, B. and O’Reilly, R.C., 2001. Interactions between frontal cortex and basal ganglia in working memory: a computational model. Cognitive, Affective, & Behavioral Neuroscience, 1, pp.137-160.

(6) Braver, T.S. and Cohen, J.D., 2000. On the control of control: The role of dopamine in regulating prefrontal function and working memory. Control of cognitive processes: Attention and performance XVIII, (2000).

(6) Page 4 - "Learned weighting w" - it is somewhat confusing to use "w" as that is commonly used for synaptic weights, whereas I understand this to be an attentional modulation vector with the same dimensionality as the grid cell code. It seems more similar to a neural network bias input than a weight matrix.

We refer to the first paragraph of our response above to the topic “Clarification of DPP-A attentional modulation” under “Major comments (Public Reviews)”, which contains our response to this issue.

(7) Page 4 - "parameterization of w... by two loss functions over the training set." - I realize that this has been stated here, but to emphasize the significance to a naïve reader, I think they should emphasize that the learning is entirely focused on the initial training space, and there is NO training done in the test spaces. It's very impressive that the parameterization is allowing generalization to translated or scaled spaces without requiring ANY training on the translated or scaled spaces.

We have added the sentence “Note that learning of parameter occurs only over the training space and is not further modified during testing (i.e. over the test spaces)” to the updated manuscript.

(8) Page 4 - "The first," - This should be specific - "The first loss function"

We have changed it to “The first loss function” in the updated manuscript.

(9) Page 4 - The analogy task seems rather simplistic when first presented (i.e. just a spatial translation to different parts of a space, which has already been shown to work in simulations of spatial behavior such as Erdem and Hasselmo, 2014 or Bush, Barry, Manson, Burgess, 2015). To make the connection to analogy, they might provide a brief mention of how this relates to the analogy space created by word2vec applied to traditional human verbal analogies (i.e. king-man+woman=queen).

We agree that the analogy task is simple, and recognize that grid cells can be used to navigate to different parts of space over which the test analogies are defined when those are explicitly specified, as shown by Erdem and Hasselmo (2014) and Bush, Barry, Manson, and Burgess (2015). However, for the analogy task, the appropriate set of grid cell embeddings must be identified that capture the same relational structure between training and test analogies to demonstrate strong OOD generalization, and that is achieved by the attentional mechanism DPP-A. As suggested by the reviewer’s comment, our analogy task is inspired by Rumelhart’s parallelogram model of analogy [1,2] (and therefore similar to traditional human verbal analogies) in as much as it involves differences (i.e A - B = C - D, where A, B, C, D are vectors in 2D space). We have now noted this in Section 2.1.1 of the updated manuscript.

(1) Rumelhart, D.E. and Abrahamson, A.A., 1973. A model for analogical reasoning. Cognitive Psychology, 5(1), pp.1-28.

(2) Mikolov, T., Chen, K., Corrado, G. and Dean, J., 2013. Efficient estimation of word representations in vector space. arXiv preprint arXiv:1301.3781.

(10) Page 5 - The variable "KM" is a bit confusing when it first appears. It would be good to re-iterate that K and M are separate points and KM is the vector between these points.

We apologize for the confusion on this point. KM is meant to refer to an integer value, obtained by multiplying K and M, which is added to both dimensions of A, B, C and D, which are points in ℤ2, to translate them to a different region of the space. K is an integer value ranging from 1 to 9 and M is also an integer value denoting the size of the training region, which in our implementation is 100. We have clarified this in Section 2.1.1 of the updated manuscript.

(11) Page 5 - "two continuous dimensions (Constantinescu et al._)" - this ought to give credit to the original study showing the abstract six-fold rotational symmetry for spatial coding (Doeller, Barry and Burgess).

We have now cited the original work by Doeller et al. [1] along with Constantinescu et al. (2016) in the updated manuscript after the phrase “two continuous dimensions” on page 5.

(1) Doeller, C.F., Barry, C. and Burgess, N., 2010. Evidence for grid cells in a human memory network. Nature, 463(7281), pp.657-661.

(12) Page 6 - Np=100. This is done later, but it would be clearer if they right away stated that Np*Nf=900 in this first presentation.

We have now added this sentence after Np=100. “Hence Np*Nf=900, which denotes the number of grid cells.”

(13) Page 6 - They provide theorem 2.1 on the determinant of the covariance matrix of the grid code, but they ought to cite this the first time this is mentioned.

We have cited Gilenwater et al. (2012) before mentioning theorem 2.1. The sentence just before that reads “We use the following theorem from Gillenwater et al. (2012) to construct :”

(14) Page 6 - It would greatly enhance the impact of the paper if they could give neuroscientists some sense of how the maximization of the determinant of the covariance matrix of the grid cell code could be implemented by a biological circuit. OR at least to show an example of the output of this algorithm when it is used as an inner product with the grid cell code. This would require plotting the grid cell code in the spatial domain rather than the 900 element vector.

We refer to our response above to the topic “Biological plausibility of DPP-A” and second, third, and fourth paragraphs of our response above to the topic “Clarification of DPP-A attentional modulation” under “Major comments (Public Reviews)”, which contain our responses to this issue.

(15) Page 6 - "That encode higher spatial frequencies..." This seems intuitive, but it would be nice to give a more intuitive description of how this is related to the determinant of the covariance matrix.

We refer to the third paragraph of our response above to the topic “Clarification of DPP-A attentional modulation” under “Major comments (Public Reviews)”, which contains our response to this issue.

(16) Page 7 - log of both sides... Nf is number of frequencies... Would be good to mention here that they are referring to equation 6 which is only mentioned later in the paragraph.

As suggested, we now refer to Equation 6 in the updated manuscript. The sentence now reads “This is achieved by maximizing the determinant of the covariance matrix over the within frequency grid cell embeddings of the training data, and Equation 6 is obtained by applying the log on both sides of Theorem 2.1, and in our case where refers to grid cells of a particular frequency.”

(17) Page 7 - Equation 6 - They should discuss how this is proposed to be implemented in brain circuits.

We refer to our response above to the topic “Biological plausibility of DPP-A” under “Major comments (Public Reviews)”, which contains our response to this issue.

1. Page 9 - "egeneralize" - presumably this is a typo?

Yes. We have corrected it to “generalize” in the updated manuscript.

(19) Page 9 - "biologically plausible encoding scheme" - This is valid for the grid cell code, but they should be clear that this is not valid for other parts of the model, or specify how other parts of the model such as DPP-A could be biologically plausible.

We refer to our response above to the topic “Biological plausibility of DPP-A” under “Major comments (Public Reviews)”, which contains our response to this issue.

(20) Page 12 - Figure 7 - comparsion to one-hots or smoothed one-hots. The text should indicate whether the smoothed one-hots are similar to place cell coding. This is the most relevant comparison of coding for those knowledgeable about biological coding schemes.

Yes, smoothed one-hots are similar to place cell coding. We now mention this in Section 5.3 of the updated manuscript.

(21) Page 12 - They could compare to a broader range of potential biological coding schemes for the overall space. This could include using coding based on the boundary vector cell coding of the space, band cell coding (one dimensional input to grid cells), or egocentric boundary cell coding.

We appreciate these useful suggestions, which we now mention as potentially valuable directions for future work in the second paragraph of Section 6 of the updated manuscript.

(22) Page 13 - "transformers are particularly instructive" - They mention this as a useful comparison, but they might discuss further why a much better function is obtained when attention is applied to the system twice (once by DPP-A and then by a transformer in the inference module).

We refer to the last paragraph of our response above to the topic “Clarification of DPP-A attentional modulation” under “Major comments (Public Reviews)”, which contains our response to this issue.

(23) Page 13 - "Section 5.1 for analogy and Section 5.2 for arithmetic" - it would be clearer if they perhaps also mentioned the specific figures (Figure 4 and Figure 6) presenting the results for the transformer rather than the LSTM.

We have now rephrased to also refer to the figures in the updated manuscript. The phrase now reads “a transformer (Figure 4 in Section 5.1 for analogy and Figure 6 in Section 5.2 for arithmetic tasks) failed to achieve the same level of OOD generalization as the network that used DPP-A.”

(24) Page 14 - "statistics of the training data" - The most exciting feature of this paper is that learning during the training space analogies can so effectively generalize to other spaces based on the right attention DPP-A, but this is not really made intuitive. Again, they should illustrate the result of the xT w inner product to demonstrate why this work so effectively!

We refer to the second, third, and fourth paragraphs of our response above to the topic “Clarification of DPP-A attentional modulation” under “Major comments (Public Reviews)”, which contains our response to this issue.

(25) Bibliography - Silver et al., go paper - journal name "nature" should be capitalized. There are other journal titles that should be capitalized. Also, I believe eLife lists family names first.

We have made the changes to the bibliography of the updated manuscript suggested by the reviewer.